# Causal Effect Identification in lvLiNGAM from Higher-Order Cumulants

**Daniele Tramontano** [1]  **Yaroslav Kivva** [2]  **Saber Salehkaleybar** [3]  **Negar Kiyavash** [2]  **Mathias Drton** [1 4]

## Abstract

This paper investigates causal effect identification in latent variable Linear Non-Gaussian Acyclic Models (lvLiNGAM) using higher-order cumulants, addressing two prominent setups that are challenging in the presence of latent confounding: (1) a single proxy variable that may causally influence the treatment and (2) underspecified instrumental variable cases where fewer instruments exist than treatments. We prove that causal effects are identifiable with a single proxy or instrument and provide corresponding estimation methods. Experimental results demonstrate the accuracy and robustness of our approaches compared to existing methods, advancing the theoretical and practical understanding of causal inference in linear systems with latent confounders.

## 1. Introduction

Predicting the impact of an unseen intervention in a system is a crucial challenge in many fields, such as medicine (Sanchez et al., 2022; Michoel & Zhang, 2023), policy evaluation (Athey & Imbens, 2017), fair decision-making (Kilbertus et al., 2017), and finance (de Prado, 2023). Randomized experiments/interventional studies are the gold standard for addressing this challenge but are often infeasible due to a variety of reasons, such as ethical concerns or prohibitively high costs. Thus, when merely observational data is available, additional assumptions on the underlying causal system are needed to compensate for the lack of interventional data. The field of causal inference seeks to formalize such assumptions. One notable approach in causal inference is modeling causal relationships through structural causal models (SCM) (Pearl, 2009). In this framework, a random vector

is associated with a directed acyclic graph (DAG). Each vector component is associated with a node in the graph and is a function of the random variables corresponding to its parents in the graph and some exogenous noise.

In general, latent confounders, i.e., unobserved variables affecting the treatment and the outcome of interest, often render the causal effect non-identifiable from the observational distribution (Shpitser & Pearl, 2006). However, in some cases and under further assumptions on the causal mechanisms, the causal effect may still be identifiable from observational data (Barber et al., 2022).

Linear models are among the most well-studied mechanisms and serve as a foundational abstraction in many scientific disciplines because they offer simple qualitative interpretations and can be learned with moderate sample sizes (Pe'er & Hacohen, 2011, Principle 1). When the exogenous noises in a linear SCM are Gaussian, the entire distributional information is contained in the variables' covariance matrix. Consequently, the higher-order cumulants of the distribution are uninformative (Marcinkiewicz, 1939, Thm. 2). As a result, the causal structure and other causal quantities are often not identifiable from mere observational data. For instance, in the context of causal structure learning, this means the causal graph is identifiable only up to an equivalence class (e.g., Drton, 2018, §10). This motivated the widespread use of the linear non-Gaussian acyclic model (LiNGAM).

The seminal work of Shimizu et al. (2006) showed that in the setting of LiNGAM, the true underlying causal graph is uniquely identifiable when all the variables are observed. Since then, a rich literature on this topic has emerged, focusing mainly on the identification and the estimation of the causal graph; see, e.g., Adams et al. (2021); Shimizu (2022); Yang et al. (2022); Wang et al. (2023); Wang & Drton (2023) for recent results that allow for the presence of hidden variables.

Within the LiNGAM literature, causal effect identification has received less attention; a complete characterization of the identifiable causal effects was provided only recently by Tramontano et al. (2024b). The drawback of this characterization is that it is based on solving an overcomplete independent component analysis (OICA) problem, known to be non-separable (Eriksson & Koivunen, 2004). Hence, the approach of Tramontano et al. (2024b) does not translate

[1]Technical University of Munich, Munich, Germany [2]Ecole Polytechnique Fédérale de Lausanne, Lausanne, Switzerland [3]Leiden Institute of Advanced Computer Science, Leiden University, Netherlands [4]Munich Center for Machine Learning, Munich, Germany. Correspondence to: Daniele Tramontano <daniele.tramontano@tum.de>.

*Proceedings of the $42^{nd}$ International Conference on Machine Learning*, Vancouver, Canada. PMLR 267, 2025. Copyright 2025 by the author(s).

into a consistent estimation method for identifiable causal effects (Tramontano et al., 2024b, §5.3).

Recent works (Kivva et al., 2023; Shuai et al., 2023) have exploited non-Gaussianity by utilizing higher-order moments to derive estimation formulas for causal effects in specific causal graphs, avoiding reliance on the challenging OICA problem. A notable scenario involves the use of a proxy variable for the latent confounder (Tchetgen et al., 2024). In LiNGAM, causal effects are identifiable from higher-order moments if every latent confounder has a corresponding proxy variable, and no proxy directly influences either the treatment or the outcome (Kivva et al., 2023). However, the method in Kivva et al. (2023) fails to produce consistent estimates when these assumptions are violated. Another important setup arises when an instrumental variable affects the outcome solely through the treatment (Angrist & Pischke, 2009, §4). For linear models, two-stage least squares (TSLS) regression can estimate causal effects when there is at least one valid instrument per treatment (Angrist & Pischke, 2009, §3.2). However, TSLS is based only on the covariance matrix, and in cases where the number of instruments is fewer than the number of treatments—referred to as underspecified instrumental variables—causal effects are not identifiable from the covariance matrix alone. This underspecification is often encountered in biological applications (Ailer et al., 2023; 2024).

This paper advances the field by providing identifiability results for causal effects using higher-order cumulants in two challenging setups: (1) a single proxy variable that may causally influence the treatment and (2) underspecified instrumental variables.

## 1.1. Contribution

Our first main contributions are identifiability results for the causal effects of interest in the aforementioned setups.

1. In the proxy variable setup (Section 3.1), unlike previous work, our proposed method allows a causal edge from the proxy to the treatment. Additionally, it recovers the causal effect for any $l$ latent confounders using a single proxy variable, in contrast to Kivva et al. (2023, Alg. 1), which requires one proxy variable per latent confounder. Furthermore, we prove that for the proxy variable graph in Fig. 3, identification from the second and third-order cumulants alone is not possible.

2. In the underspecified instrumental variable setup (Section 3.2), we demonstrate that the causal effects of multiple treatments can be identified using only a single instrumental variable. This relaxes the requirement in the existing literature on linear instrumental variables, which traditionally assumes the number of instruments to be greater than or equal to the number of treatments.

Our second main contribution consists of practical methods to estimate identifiable causal effects in both considered setups. The methods build on the identifiability results and process finite-sample estimates of higher-order cumulants (Section 4). Our experiments show that the proposed approach provides consistent estimators in causal graphs, for which previous methods in the literature fail (Section 6).

## 2. Problem Definition

### 2.1. Notation

A *directed graph* is a pair $\mathcal{G} = (\mathcal{V}, E)$ where $\mathcal{V} = [p] := \{1, \ldots, p\}$ is the set of nodes and $E \subseteq \{(i, j) \mid i, j \in \mathcal{V}, i \neq j\}$ is the set of edges. We denote a pair $(i, j) \in E$ as $i \to j$.

A (directed) path from node $i$ to node $j$ in $\mathcal{G}$ is a sequence of nodes $\pi = (i_1 = i, \ldots, i_{k+1} = j)$ such that $i_s \to i_{s+1} \in E$ for $s \in \{1, \ldots, k\}$. A cycle in $\mathcal{G}$ is a path from a node $i$ to itself. A *Directed Acyclic Graph* (DAG) is a directed graph without cycles. If $i \to j \in E$, we say that $i$ is a parent of $j$, and $j$ is a child of $i$. If there is a path from $i$ to $j$ in $\mathcal{G}$, we say that $i$ is an ancestor of $j$ and $j$ is a descendant of $i$. The sets of parents, children, ancestors, and descendants of a given node $i$ are denoted by $\mathrm{pa}(i), \mathrm{ch}(i), \mathrm{an}(i)$, and $\mathrm{de}(i)$, respectively. In our work, we distinguish between observed and latent variables by partitioning the nodes into two sets $\mathcal{V} = \mathcal{O} \cup \mathcal{L}$, of respective sizes $p_o$ and $p_l$. We write tensors in boldface. The entry $(i_1, \ldots, i_k)$ of a tensor $\mathbf{T}$ is denoted by $\mathbf{t}_{i_1, \ldots, i_k}$.

Cumulants are alternative representations of moments of a distribution that are particularly useful when dealing with linear SCM (Robeva & Seby, 2021). Here, we formalize the definition and discuss their basic properties.

**Definition 2.1.** The $k$-th cumulant tensor of a random vector $\mathbf{N} = [N_1, \ldots, N_p]$ is the $k$-way tensor in $\mathbb{R}^{p \times \cdots \times p} \equiv (\mathbb{R}^p)^k$ whose entry in position $(i_1, \ldots, i_k)$ is the cumulant

$$\mathbf{c}^{(k)}(\mathbf{N})_{i_1, \ldots, i_k} :=$$

$$\sum_{(A_1, \ldots, A_L)} (-1)^{L-1} (L-1)! \mathbb{E}\left[\prod_{j \in A_1} N_j\right] \cdots \mathbb{E}\left[\prod_{j \in A_L} N_j\right],$$

where the sum is taken over all partitions $(A_1, \ldots, A_L)$ of the multiset $\{i_1, \ldots, i_k\}$.

Cumulant tensors are symmetric, i.e.,

$$\mathbf{c}^{(k)}(\mathbf{N})_{i_1, \ldots, i_k} = \sigma(\mathbf{c}^{(k)}(\mathbf{N}))_{i_1, \ldots, i_k}$$
$$:= \mathbf{c}^{(k)}(\mathbf{N})_{\sigma(i_1), \ldots, \sigma(i_k)} \ \forall \sigma \in S_k,$$

where $S_k$ is the symmetric group on $[k]$. We write $\mathrm{Sym}_k(p)$ for the subspace of symmetric tensors in $(\mathbb{R}^p)^k$.

**Lemma 2.2** (Comon & Jutten, 2010, §5)**.** *If the entries of* $\mathbf{N} = [N_1, \ldots, N_p]$ *are jointly independent, then* $\mathbf{c}^{(k)}(\mathbf{N})$ *is*

diagonal, i.e., $\mathbf{c}^{(k)}(\mathbf{N})_{i_1,\ldots,i_k}$ is equal to 0 unless $i_1 = i_2 = \cdots = i_k = i$, for some $i \in [p]$.

We write $\mathrm{Diag}^k(p)$ for the space of order $k$ diagonal tensors.

**Lemma 2.3** (Comon & Jutten, 2010, §5). *Let* $\mathbf{N} = [N_1, \ldots, N_p]$ *be any p-variate random vector, and* $\mathbf{A} \in \mathbb{R}^{s \times p}$ *for any* $s \in \mathbb{N}$, *then*

$$\mathbf{c}^{(k)}(\mathbf{A} \cdot \mathbf{N})_{i_1,\ldots,i_k} = \sum_{1 \le j_1,\ldots,j_k \le p} \mathbf{c}^{(k)}(\mathbf{N})_{j_1,\ldots,j_k} \mathbf{a}_{j_1,i_1} \cdots \mathbf{a}_{j_k,i_k}.$$

*In terms of the entire k-th cumulant tensor, this amounts to*

$$\mathbf{C}^{(k)}(\mathbf{A} \cdot \mathbf{N}) = \mathbf{C}^{(k)}(\mathbf{N}) \bullet_k \mathbf{A} \tag{1}$$

*where* $\bullet_k$ *is the Tucker product between* $\mathbf{C}^{(k)}(\mathbf{N})$ *and* $\mathbf{A}$.

### 2.2. Model

Let $\mathcal{G} = (\mathcal{V}, E)$ be a *fixed* DAG on $p$ nodes. On a fixed probability space, let $\mathbf{V} = [V_0, \ldots, V_p]$ be a random vector taking values in $\mathbb{R}^p$ and satisfying the following SCM:

$$\mathbf{V} = \mathbf{A}\mathbf{V} + \mathbf{N} = \mathbf{B}\mathbf{N}, \tag{2}$$

where $\mathbf{a}_{j,i} = 0$ if $i \to j \notin E$, matrix $\mathbf{B} := (\mathbf{I} - \mathbf{A})^{-1}$, and the entries of the exogenous noise vector $\mathbf{N}$ are assumed to be jointly independent and *non-Gaussian*. $\mathbf{V}$ is partitioned into $[\mathbf{V}_o, \mathbf{V}_l]$, where $\mathbf{V}_o$ is observed of dimension $p_o$, while $\mathbf{V}_l$ is latent and of dimension $p_l$. We can rewrite (2) as

$$\begin{bmatrix} \mathbf{V}_o \\ \mathbf{V}_l \end{bmatrix} = \begin{bmatrix} \mathbf{A}_{o,o} & \mathbf{A}_{o,l} \\ \mathbf{A}_{l,o} & \mathbf{A}_{l,l} \end{bmatrix} \begin{bmatrix} \mathbf{V}_o \\ \mathbf{V}_l \end{bmatrix} + \begin{bmatrix} \mathbf{N}_o \\ \mathbf{N}_l \end{bmatrix},$$

which implies that the observed random vector satisfies

$$\mathbf{V}_o = \mathbf{B}'\mathbf{N} = \begin{bmatrix} \mathbf{B}_o & \mathbf{B}_l \end{bmatrix} \begin{bmatrix} \mathbf{N}_o \\ \mathbf{N}_l \end{bmatrix}, \tag{3}$$

where $\mathbf{B}' := [(\mathbf{I} - \mathbf{A})^{-1}]_{\mathcal{O},\mathcal{V}}$ is known as the mixing matrix. This model for $\mathbf{V}_o$ is known as the latent variable LiNGAM (lvLiNGAM).

Salehkaleybar et al. (2020, §3) showed that the two parts of the matrix $\mathbf{B}'$ can be expressed as follows:

$$\mathbf{B}_o = (\mathbf{I} - \mathbf{A}')^{-1}, \quad \mathbf{B}_l = (\mathbf{I} - \mathbf{A}')^{-1}\mathbf{A}_{o,l}(\mathbf{I} - \mathbf{A}_{l,l})^{-1},$$

with $\mathbf{A}' = \mathbf{A}_{o,o} + \mathbf{A}_{o,l}(\mathbf{I} - \mathbf{A}_{l,l})^{-1}\mathbf{A}_{l,o}$. The matrix $\mathbf{B}' = (\mathbf{b}'_{i,j})$ contains information on the interventional distributions of $\mathbf{V}_o$. In particular,[1]

$$\mathbf{b}'_{i,j} = \frac{\partial \mathbb{E}(V_i \mid \mathrm{do}(V_j))}{\partial V_j},$$

---

[1]See Pearl (2009, §3) for the definition of *do* intervention.

i.e., $\mathbf{b}'_{i,j}$ is the average total causal effect of $j$ on $i$.

Hoyer et al. (2008) showed that for any lvLiNGAM model, an associated *canonical model* exists, in which, in the corresponding graph, all the latent nodes have at least two children and have no parents. We refer to the graph corresponding to a canonical model as a canonical graph. The original and the associated canonical model are *observationally* and *causally* equivalent (Hoyer et al., 2008, §3). Subsequently, without loss of generality, we will assume our model is canonical in this sense.

In canonical models, $\mathbf{A}_{l,o} = \mathbf{A}_{l,l} = \mathbf{0}$, and in particular

$$\mathbf{B}_o = (\mathbf{I} - \mathbf{A}_{o,o})^{-1}, \qquad \mathbf{B}_l = (\mathbf{I} - \mathbf{A}_{o,o})^{-1}\mathbf{A}_{o,l}. \tag{4}$$

For every canonical $\mathcal{G}$, let $\mathbb{R}^{\mathcal{G}}_{\mathbf{A}}$ be the set of all $p \times p$ real matrices $\mathbf{A}$ such that $\mathbf{a}_{i,j} = 0$ if $j \to i \notin \mathcal{G}$. Let $\mathbb{R}^{\mathcal{G}} \subset \mathbb{R}^{p_o \times p}$ be the set of all matrices $\mathbf{B}' = [\mathbf{B}_o, \mathbf{B}_l]$ that can be obtained from a matrix $\mathbf{A} \in \mathbb{R}^{\mathcal{G}}_{\mathbf{A}}$ according to (4). Let $\mathrm{NG}^p$ be the set of $p$ dimensional, non-degenerate, jointly independent *non-Gaussian* random vectors, and let $\mathcal{M}(\mathcal{G})$ be the set of all $p_o$ dimensional random vectors that can be expressed according to (3) with $\mathbf{B}' \in \mathbb{R}^{\mathcal{G}}$. Moreover, we define $\mathcal{M}^{(k)}(\mathcal{G}) \subseteq \mathrm{Sym}_k(p_o)$ to be the set of symmetric $k$-th tensors that can be obtained as $k$-cumulant tensor for distributions in $\mathcal{M}(\mathcal{G})$, i.e.,

$$\mathcal{M}^{(k)}(\mathcal{G}) := \{\mathbf{C}^{(k)}(\mathbf{V}_o) \mid \mathbf{V}_o \in \mathcal{M}(\mathcal{G})\} = \{\mathbf{D}^{(k)} \bullet_k \mathbf{B}' \mid \mathbf{D}^{(k)} \in \mathrm{Diag}^k(p), \mathbf{B}' \in \mathbb{R}^{\mathcal{G}}\},$$

where the set-equality is due to Lemma 2.3. Using the second equality, we can define the following polynomial parameterization for $\mathcal{M}^{(k)}(\mathcal{G})$:

$$\begin{aligned} \Phi^{(k)}_{\mathcal{G}} : \mathbb{R}^{\mathcal{G}} \times \mathrm{Diag}^k(p) &\to \mathcal{M}^{(k)}(\mathcal{G}) \\ (\mathbf{B}', \mathbf{D}^{(k)}) &\mapsto \mathbf{D}^{(k)} \bullet_k \mathbf{B}'. \end{aligned} \tag{5}$$

This map expresses the tensor of observed cumulants in terms of the tensor of exogenous cumulants and the mixing matrix. Finally, we define $\mathcal{M}^{(\le k)}(\mathcal{G}) := \mathcal{M}^{(2)}(\mathcal{G}) \times \cdots \times \mathcal{M}^{(k)}(\mathcal{G})$, and similarly $\mathrm{Diag}^{(\le k)}(p)$ and $\Phi^{(\le k)}_{\mathcal{G}}$.

### 2.3. Identifiability

In this work, we are interested in identifying specific entries of the mixing matrix from finitely many cumulants of the observational distribution. We formalize the problem as follows. We say that the causal effect from $j$ to $i$ is *generically* identifiable from the first $k$ cumulants of the distribution if there is a Lebesgue measure zero subset $\mathcal{S}^{\mathcal{G}}_k$ of $\mathbb{R}^{\mathcal{G}} \times \mathrm{Diag}^{(\le k)}(p)$ such that for all $(\mathbf{B}', \mathbf{D}^{(\le k)}) \in (\mathbb{R}^{\mathcal{G}} \times \mathbf{D}^{(\le k)}) \setminus \mathcal{S}^{\mathcal{G}}_k$, we have $\mathbf{b}'_{i,j} = \tilde{\mathbf{b}}'_{i,j}$ for every other mixing matrix $\tilde{\mathbf{B}}' \in \mathbb{R}^{\mathcal{G}}$ that can define the same cumulants up to order $k$, that is, whenever $\Phi^{(\le k)}_{\mathcal{G}}(\tilde{\mathbf{B}}', \tilde{\mathbf{D}}^{(\le k)}) = \Phi^{(\le k)}_{\mathcal{G}}(\mathbf{B}', \mathbf{D}^{(\le k)})$ for some $\tilde{\mathbf{D}}^{(\le k)} \in \mathrm{Diag}^{(\le k)}(p)$.

For the remainder of the text, whenever we use the term generic, it is implied that the result holds outside the Lesbegue measure zero subset of the parameter space $\mathcal{S}_k^{\mathcal{G}}$.

*Remark* 2.4 (The scaling matrix). Equation (4) implies that as long as we are focused on identifying the causal effect between observed variables alone, the scaling of the latent columns does not make a difference. Hence, without loss of generality, we assume subsequently that all mixing matrices are scaled so that the first non-zero entry in each column is equal to 1. In other words, $\mathbf{a}_{i,l} = 1$ if $i$ is the first child of $l$ in a given causal order, where $i$ and $l$ are observed and latent variables, respectively.

## 3. Main Results

This section presents our main identifiability results. Section 3.1 treats the case of a proxy variable. Section 3.2 details our findings for underspecified instrumental variables case.

Before presenting our results, we review two key results from Schkoda et al. (2024) pertaining to the causal graph $\mathcal{G}^l$ depicted in Fig. 1, which includes two observed variables, $V_1$ and $V_2$, along with $l$ latent variables $L_1, \ldots, L_l$. They will be used to establish our identifiability results.

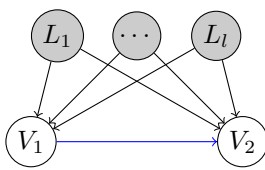

*Figure 1.* The causal graph $\mathcal{G}^l$ with $l$ latent confounders.

**Theorem 3.1** (Schkoda et al., 2024, Thm. 4). *Consider the causal graph $\mathcal{G}^l$ with two observed variables and $l$ latent variables depicted in Fig. 1. There is a polynomial of degree $l + 1$ with coefficients expressed in terms of the first $k(l) := (l+2) + \lceil(-3+\sqrt{8l+17})/2\rceil$ cumulants of the distributions where the roots of the polynomial are $\mathbf{b}_{2,1}, \mathbf{b}_{2,L_1}, \ldots, \mathbf{b}_{2,L_l}$. We refer to this polynomial as $p_{\mathbf{V},l}(\mathbf{b})$ (see Remark B.1 in the appendix for a definition of the polynomial).*

The above theorem implies that in the causal graph $\mathcal{G}^l$, one can identify the causal effect of interest, $\mathbf{b}_{2,1}$, up to a set of size $l + 1$ using the first $k(l)$ cumulants of the distribution. In Section 3.1 (and Section 3.2), we demonstrate how incorporating a proxy (or instrumental) variable can refine this result, enabling unique identification of the causal effect. This approach involves deriving additional polynomial equations among the cumulants of the observed distribution, for which the true causal effect is a solution.

**Example 3.2** (Polynomial for the graph in Fig. 1). *For the special case $l = 1$, the polynomial equation described in Theorem 3.1 is defined as follows with the coefficients*

*expressed in terms of first $k(l = 1) = 4$:*

$$\mathbf{b}^2 \cdot (\mathbf{c}(\mathbf{V})_{1,1,1,2}\mathbf{c}(\mathbf{V})_{1,1,2} - \mathbf{c}(\mathbf{V})_{1,1,2,2}\mathbf{c}(\mathbf{V})_{1,1,1})$$
$$+ \mathbf{b} \cdot (\mathbf{c}(\mathbf{V})_{1,2,2,2}\mathbf{c}(\mathbf{V})_{1,1,1} - \mathbf{c}(\mathbf{V})_{1,1,1,2}\mathbf{c}(\mathbf{V})_{1,2,2})$$
$$- (\mathbf{c}(\mathbf{V})_{1,2,2,2}\mathbf{c}(\mathbf{V})_{1,1,2} + \mathbf{c}(\mathbf{V})_{1,1,2,2}\mathbf{c}(\mathbf{V})_{1,2,2}) = 0. \tag{6}$$

**Lemma 3.3** (Schkoda et al., 2024, Lemma 5). *Consider the causal graph $\mathcal{G}^l$ from Fig. 1. For every integer $k \geq 2$, the exogenous cumulant vector $[\mathbf{c}^k(\mathbf{N})_{1,\ldots,1}, \mathbf{c}^k(\mathbf{N})_{L_1,\ldots,L_1}, \ldots, \mathbf{c}^k(\mathbf{N})_{L_l,\ldots,L_l}]$ is a solution of the following linear system*

$$\begin{bmatrix} 1 & 1 & \cdots & 1 \\ \mathbf{b}_{2,1} & \mathbf{b}_{2,L_1} & \cdots & \mathbf{b}_{2,L_l} \\ \vdots & \vdots & \ddots & \vdots \\ \mathbf{b}_{2,1}^{k-1} & \mathbf{b}_{2,L_1}^{k-1} & \cdots & \mathbf{b}_{2,L_l}^{k-1} \end{bmatrix} \begin{bmatrix} \mathbf{c}^k(\mathbf{N})_{1,\ldots,1} \\ \mathbf{c}^k(\mathbf{N})_{L_1,\ldots,L_1} \\ \vdots \\ \mathbf{c}^k(\mathbf{N})_{L_l,\ldots,L_l} \end{bmatrix}$$
$$= \begin{bmatrix} \mathbf{c}^k(\mathbf{V}_o)_{1,\ldots,1} \\ \mathbf{c}^k(\mathbf{V}_o)_{1,\ldots,1,2} \\ \vdots \\ \mathbf{c}^k(\mathbf{V}_o)_{1,2\ldots,2} \end{bmatrix}. \tag{7}$$

*The solution is, generically, unique if $k \geq l + 1$.*

Let $\mathbf{b}$ be the vector $[\mathbf{b}_{2,1}, \mathbf{b}_{2,L_1}, \cdots, \mathbf{b}_{2,L_l}]$. We rewrite the system in (7) as

$$\mathrm{M}(\mathbf{b}, k) \cdot \mathbf{c}^k = \mathbf{c}_{(1,2)}^k(\mathbf{V}_o). \tag{8}$$

The above lemma implies that after using Theorem 3.1 to recover $[\mathbf{b}_{21}, \mathbf{b}_{2L_1}, \ldots, \mathbf{b}_{2L_l}]$ up to a permutation, it is possible to estimate some cumulants corresponding to the exogenous noises of $V_1$ and the $l$ latent variables up to the same permutation.

### 3.1. Proxy Variable

In this section, we first provide the identifiability result for a causal graph with a single proxy variable and $l$ latent variables where there is no edge from the proxy variable to the treatment. Then, we extend our result to the case where there is an edge from the proxy to the treatment.

#### 3.1.1. NO EDGE FROM PROXY TO TREATMENT

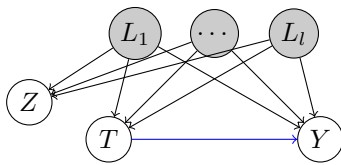

*Figure 2.* The causal graph with a single proxy variable $Z$ and $l$ latent confounders $L_1, \cdots, L_l$ where there is no edge from the proxy to the treatment.

**Theorem 3.4.** *In the lvLiNGAM for the causal graph in Fig. 2, with the proxy variable $Z$ and $l$ latent confounders $L_1, \ldots, L_l$, the causal effect from $T$ to $Y$ is generically identifiable from the first $k(l)$ cumulants of the observational distribution.*

*Proof.* Considering the pairs $[Z, T]$, $[Z, Y]$, and $[T, Y]$ as pair $[V_1, V_2]$ in Theorem 3.1, we obtain the vectors

$$
\begin{aligned}
\mathbf{b}^{ZT} &= [0, \mathbf{b}_{T,L_1}, \ldots, \mathbf{b}_{T,L_l}], \\
\mathbf{b}^{ZY} &= [0, \mathbf{b}_{Y,L_1}, \ldots, \mathbf{b}_{Y,L_l}], \\
\mathbf{b}^{TY} &= [\mathbf{b}_{Y,T}, \mathbf{b}_{Y,L_1}/\mathbf{b}_{T,L_1}, \ldots, \mathbf{b}_{Y,L_l}/\mathbf{b}_{T,L_l}],
\end{aligned}
\tag{9}
$$

up to some permutations (notice that the ratios in the last equation are a consequence of the choice of the scaling we discussed in Remark 2.4). Next, we recover the vector

$$
[\mathbf{c}^{l+1}(\mathbf{N})_{1,\ldots,1}, \mathbf{c}^{l+1}(\mathbf{N})_{L_1,\ldots,L_1}, \ldots, \mathbf{c}^{l+1}(\mathbf{N})_{L_l,\ldots,L_l}] \tag{10}
$$

using Lemma 3.3 twice (up to some permutations) with the vector $\mathbf{b}^{ZT}$, and then with $\mathbf{b}^{ZY}$ by solving the linear system in (7). Since the cumulants of different exogenous noises are generically distinct, we can match the entries in $\mathbf{b}^{ZT}$ to their corresponding entries in $\mathbf{b}^{ZY}$ using the two recovered exogenous cumulant vectors. This allows us to construct a new vector

$$
\mathbf{b}^r := \left[ \mathbf{b}_{Y,L_1}/\mathbf{b}_{T,L_1}, \ldots, \mathbf{b}_{Y,L_l}/\mathbf{b}_{T,L_l} \right]. \tag{11}
$$

Finally, $\mathbf{b}_{Y,T}$ is the only entry in $\mathbf{b}^{TY}$ that does not equal any entry of $\mathbf{b}^r$. $\square$

### 3.1.2. WITH AN EDGE FROM PROXY TO TREATMENT

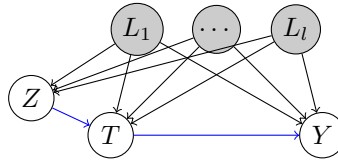

*Figure 3.* The causal graph with a single proxy variable $Z$ and $l$ latent confounders $L_1, \cdots, L_l$ where there is an edge from the proxy to the treatment.

**Theorem 3.5.** *In the lvLiNGAM for the causal graph in Fig. 3, the causal effect from $T$ to $Y$ is generically identifiable from the first $k(l)$ cumulants of the observational distribution.*

*Proof.* Let $\mathbf{b}$ be either equal to $[\mathbf{b}_{T,Z}, \mathbf{b}_{Y,Z}]$ or to $[\mathbf{b}_{T,L_i}, \mathbf{b}_{Y,L_i}]$ for some $i \in [l]$. Then, the triple

$$
\mathbf{V}^{\mathbf{b}} := [Z, T - \mathbf{b}_1 Z, Y - \mathbf{b}_2 Z] \tag{12}
$$

follows a lvLiNGAM model compatible with the graph in Fig. 2 with the causal effect from $T - \mathbf{b}_1 Z$ to $Y - \mathbf{b}_2 Z$ being

the same as in the original model (see Lemma B.2). Hence, once we have one of these pairs, we can use Theorem 3.4 to recover the causal effects between $T$ and $Y$.

To obtain the pairs, we apply Theorem 3.1 to $[Z, T]$ and $[Z, Y]$, finding

$$
\begin{aligned}
\mathbf{b}^T &= [\mathbf{b}_{T,Z}, \mathbf{b}_{T,L_1}, \ldots, \mathbf{b}_{T,L_l}], \\
\mathbf{b}^Y &= [\mathbf{b}_{Y,Z}, \mathbf{b}_{Y,L_1}, \ldots, \mathbf{b}_{Y,L_l}]
\end{aligned}
\tag{13}
$$

up to some permutations of their entries. Moreover, using Lemma 3.3, we can align the pairs of solutions as we did in the proof of Theorem 3.4. In this manner, we obtain

$$
\mathbf{b}^1 = [\mathbf{b}_{T,Z}, \mathbf{b}_{Y,Z}], \ldots, \mathbf{b}^{l+1} = [\mathbf{b}_{T,L_l}, \mathbf{b}_{Y,L_l}]. \tag{14}
$$

Any $\mathbf{b}^i$ allows us to identify the correct causal effect. $\square$

The above result shows that estimating the first $k(l)$ cumulants of the distribution is sufficient to identify the causal effect. However, since estimating higher-order cumulants is statistically more challenging, it is important to understand whether the same result can be obtained with lower-order cumulants. The next result shows that this is not possible for the case $l = 1$.

**Theorem 3.6.** *Consider the causal graph depicted in Fig. 3 with $l = 1$. Then, the causal effect from $T$ to $Y$ is not identifiable from the first $k(l) - 1 = 3$ cumulants of the observational distribution.*

*Proof.* Garcia et al. (2010, Prop. 3, 4) prove that, once a polynomial parametrization for a statistical model is known, the generic identifiability of any parameter can be verified through a Gröbner basis computation. We leveraged this fact as follows: we parameterize the model $\mathcal{M}^{(\leq 3)}(\mathcal{G})$ using (5) and compute the vanishing ideal for the modified parametrization

$$
\begin{aligned}
\tilde{\Phi}_{\mathcal{G}}^{(\leq k)} : \mathbb{R}^{\mathcal{G}} \times \mathrm{Diag}^{\leq k}(p) &\to \mathbb{R} \times \mathcal{M}^{(\leq k)}(\mathcal{G}) \\
(\mathbf{B}', \mathbf{D}^{(2)}, \mathbf{D}^{(3)}) &\mapsto [\mathbf{b}_{Y,T}, \mathbf{D}^{(2)} \bullet_2 \mathbf{B}', \mathbf{D}^{(3)} \bullet_3 \mathbf{B}'].
\end{aligned}
$$

Specifically, computing the reduced Gröbner basis for an elimination term order (see Definition A.3), we find that $\mathbf{b}_{Y,T}$ is determined merely as a root of a degree two polynomial.[2] Since $\mathbf{b}_{Y,T}$ is unconstrained in $\mathbb{R}^{\mathcal{G}}$, it is not generically identifiable (Garcia et al., 2010, Prop. 3). $\square$

### 3.2. Underspecified Instrumental Variable

We now prove that in lvLiNGAM models, one valid instrument suffices to estimate the causal effects of multiple treatments.

---

[2]The computations were done using the computer algebra software Macaulay 2 (Grayson & Stillman, 2023). The code to replicate the computation can be found at https://github.com/danieletramontano/CEId-from-Moments/blob/main/Macaulay2/NonGaussianIdentifiability.m2.

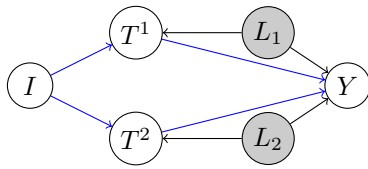

Figure 4. An example of a causal graph for the underspecified instrumental variable model.

In a causal graph $\mathcal{G}$, we say that $I$ is a valid instrument for the treatments $T^1, \ldots, T^k$ on $Y$ if

$$I \in \mathrm{pa}(T^i) \qquad \forall i \in [n],$$
$$\mathrm{an}(I) \cap \mathrm{an}(T^i) \cap \mathcal{L} = \emptyset \qquad \forall i \in [n],$$
$$I \perp_{\mathcal{G}_{\setminus T}} Y,$$

where $\perp$ denotes d-separation (Pearl, 2009, §1.2), and $\mathcal{G}_{\setminus T}$ is the graph obtained by removing the edges $T^i \rightarrow Y$ from $\mathcal{G}$ for all $i \in [n]$ (Ailer et al., 2023, Eq. 1). Fig. 4 illustrates an example with two treatments and one instrumental variable.

**Theorem 3.7.** *In the lvLiNGAM for the causal graph in Fig. 4, with instrumental variable $I$, treatments $T^1, \ldots, T^k$, and outcome $Y$, the causal effect from $T^i$ to $Y$ is generically identifiable from the first $k(l)$ cumulants of the observational distribution, where $l := \max_{i \in [n]} |\mathrm{an}(T^i) \cap \mathrm{an}(Y) \setminus I|$.*

The proof of the above result can be found in Appendix B. In the next example, we outline the identification strategy for the graph in Fig. 4.

**Example 3.8** (Identification equations for the graph in Fig. 4). *First, compute $\mathbf{b}_{T^i,I} = \mathbf{c}_{T^i,I}^2 / \mathbf{c}_{I,I}^2$ and $\mathbf{b}_{Y,I} = \mathbf{c}_{Y,I}^2 / \mathbf{c}_{I,I}^2$. Then, consider the vector*

$$\mathbf{V}^I := [T^1 - \mathbf{b}_{T^1,I}I, T^2 - \mathbf{b}_{T^2,I}I, Y - \mathbf{b}_{Y,I}I].$$

*The vector of causal effects $[\mathbf{b}_{Y,T^1}, \mathbf{b}_{Y,T^2}]$ is the unique solution to the following polynomial system:*

$$\mathbf{b}_{Y,T^1}^2 \left( \mathbf{c}(\mathbf{V}^I)_{1,1,1,3} \mathbf{c}(\mathbf{V}^I)_{1,1,3} - \mathbf{c}(\mathbf{V}^I)_{1,1,3,3} \mathbf{c}(\mathbf{V}^I)_{1,1,1} \right)$$
$$+ \mathbf{b}_{Y,T^1} \left( \mathbf{c}(\mathbf{V}^I)_{1,3,3,3} \mathbf{c}(\mathbf{V}^I)_{1,1,1} - \mathbf{c}(\mathbf{V}^I)_{1,1,1,3} \mathbf{c}(\mathbf{V}^I)_{1,3,3} \right)$$
$$- \left( \mathbf{c}(\mathbf{V}^I)_{1,3,3,3} \mathbf{c}(\mathbf{V}^I)_{1,1,3} + \mathbf{c}(\mathbf{V}^I)_{1,1,3,3} \mathbf{c}(\mathbf{V}^I)_{1,3,3} \right) = 0,$$

$$\mathbf{b}_{Y,T^2}^2 \left( \mathbf{c}(\mathbf{V}^I)_{2,2,2,3} \mathbf{c}(\mathbf{V}^I)_{2,2,3} - \mathbf{c}(\mathbf{V}^I)_{2,2,3,3} \mathbf{c}(\mathbf{V}^I)_{2,2,2} \right)$$
$$+ \mathbf{b}_{Y,T^2} \left( \mathbf{c}(\mathbf{V}^I)_{2,3,3,3} \mathbf{c}(\mathbf{V}^I)_{2,2,2} - \mathbf{c}(\mathbf{V}^I)_{2,2,2,3} \mathbf{c}(\mathbf{V}^I)_{2,3,3} \right)$$
$$- \left( \mathbf{c}(\mathbf{V}^I)_{2,3,3,3} \mathbf{c}(\mathbf{V}^I)_{2,2,3} + \mathbf{c}(\mathbf{V}^I)_{2,2,3,3} \mathbf{c}(\mathbf{V}^I)_{2,3,3} \right) = 0,$$

$$\mathbf{b}_{Y,I} - \mathbf{b}_{T^1,I} \mathbf{b}_{Y,T^1} - \mathbf{b}_{T^2,I} \mathbf{b}_{Y,T^2} = 0,$$

*where the first two equations are instances of (6), and the last equation can be derived by directly applying Lemma A.8 to the graph in Fig. 4.*

*Remark* 3.9 (Multiple instruments). For simplicity of notation, we stated the theorem in the most challenging context of a single instrumental variable. However, the result readily extends to cases with multiple valid instruments $I$, as long as each treatment is associated with at least one valid instrument. See Remark B.5 in the appendix for details on adapting the identification strategy to multiple instruments.

# 4. Estimation

In this section, we explain how to develop estimation techniques based on the identifiability results from the previous section. We assume access to an i.i.d sample $\mathbf{V}_n \in \mathbb{R}^{n \times p_o}$ drawn from the distribution of a random vector $\mathbf{V}_o \in \mathcal{M}(\mathcal{G})$ for a fixed graph $\mathcal{G}$. All algorithms will process unbiased estimates of the corresponding population cumulants, i.e., k-statistics (McCullagh, 1987, §4.2).

---

**Algorithm 1** Proxy Variable (Fig. 2)

---

**INPUT:** Data $\mathbf{V}_n = [Z_n, T_n, Y_n]$, bound on the number of latent variables $l$.

1: $\mathbf{b}_n^{ZT} \leftarrow$ roots of $p_{[Z_n,T_n],l-1}(\mathbf{b}) = 0$ {(9)}
2: $\mathbf{b}_{n,0}^{ZT} \leftarrow [0, \mathbf{b}_n^{ZT}]$
3: $\mathbf{b}_n^{ZY} \leftarrow$ roots of $p_{[Z_n,Y_n],l-1}(\mathbf{b}) = 0$ {(9)}
4: $\mathbf{b}_{n,0}^{ZY} \leftarrow [0, \mathbf{b}_n^{ZY}]$
5: $\mathbf{b}_n^{TY} \leftarrow$ roots of $p_{[T_n,Y_n],l}(\mathbf{b}) = 0$ {(9)}
6: $\mathbf{c}_{T_n}^{l+1} \leftarrow$ solution to the linear system
$\quad \mathrm{M}(\mathbf{b}_{n,0}^{ZT}, l+1) \cdot \mathbf{c}^{l+1} = \mathbf{c}_{(1,2)}^{l+1}([Z_n, T_n])$ {(8)}
7: $\mathbf{c}_{Y_n}^{l+1} \leftarrow$ solution to the linear system
$\quad \mathrm{M}(\mathbf{b}_{n,0}^{ZY}, l+1) \cdot \mathbf{c}^{l+1} = \mathbf{c}_{(1,2)}^{l+1}([Z_n, Y_n])$ {(8)}
8: $\sigma_n \leftarrow \arg\min_{\sigma \in S_{l+1}} (\|\mathbf{c}_{T_n}^{l+1} - \sigma(\mathbf{c}_{Y_n}^{l+1})\|_2^2)$
9: $\mathbf{b}_n^r \leftarrow \mathbf{b}_{n,0}^{ZT} / \sigma_n(\mathbf{b}_{n,0}^{ZY})$ {Under the convention $0/0 = 0$.}
10: $\eta_n \leftarrow \arg\min_{\eta \in S_{l+1}} (\|\mathbf{b}_n^r - \eta(\mathbf{b}_n^{TY})\|_2^2)$
11: **RETURN:** $\mathbf{b}_n^{TY}[\eta_n(1)]$

---

Algorithm 1 outlines the estimation procedure for the causal effect for the graph in Fig. 2. This algorithm replaces the steps in the proof of Theorem 3.4 with their respective finite-sample versions. Specifically, lines 1 to 5 correspond to (9), where the $l-1$ in lines 1 and 3 results from the fact that, without an edge from $Z$ to $T$, one of the roots of $p_{[Z_n,T_n],l}$ is known to be zero (Schkoda et al., 2024, Thm. 3). Lines 6 and 7 correspond to (10), and lines 7 and 8 correspond to (11). In particular, in line 8, we determine the permutation $\sigma \in S_{l+1}$ that minimizes the $\ell_2$ distance between $\mathbf{c}_{T_n}^{l+1}$ and $\sigma(\mathbf{c}_{Y_n}^{l+1})$. This step is necessary because, due to estimation error, we cannot perfectly align the entries of $\mathbf{c}_{T_n}^{l+1}$ and $\mathbf{c}_{Y_n}^{l+1}$. Similarly, in line 9, we identify the permutation $\eta(\mathbf{c}_{Y_n}^{l+1})$ that minimizes the $\ell_2$ distance between $\mathbf{b}_n^r$ and $\eta(\mathbf{b}_n^{TY})$. Finally, we return the entry of $\mathbf{b}_n^{TY}$ corresponding to the zero in $\mathbf{b}_n^r$.

The algorithms for the other graphs can be found in Ap-

pendix C. Furthermore, in Algorithm 4, we propose an optimization technique that improves the finite-sample performance for the graph in Fig. 3 with a single latent variable (as shown in the right panel of Fig. 6).

## 5. Related Work

There is a substantial body of work on causal effect identification in linear SCMs, with several graphical criteria developed for identification in a fixed causal graph. For Gaussian models, Drton et al. (2011); Kumor et al. (2020); Barber et al. (2022) provided conditions under which causal effects can be identified solely from the covariance matrix. In the non-Gaussian case, analogous results have been established by Tramontano et al. (2024a;b), with criteria that are both sound and complete but which require access to the full observational distribution.

Results for the identification of the mixing matrix (i.e., without assuming knowledge of the causal graph) are provided in Salehkaleybar et al. (2020); Yang et al. (2022); Adams et al. (2021) and in Cai et al. (2023); Schkoda et al. (2024); Chen et al. (2024); Li et al. (2025). The former results are based on solving an OICA problem (hence, are not equipped with consistent estimation methods), and the latter results, similar to our approach, rely on explicit cumulant/moment equations. Notably, both Cai et al. (2023) and Chen et al. (2024) assume specific structural conditions—namely, a *One-Latent-Component structure* and a *homologous surrogate*, respectively—which do not apply to the graphs considered in Sections 3.1 and 3.2.

In the context of proximal causal inference, Kuroki & Pearl (2014) explored two scenarios for determining causal effects: (1) discrete finite variables $Z$ and $L$: It was shown that the causal effect can be identified if $\mathbb{P}(Z|L)$ is known (e.g., from external studies) or an additional proxy variable ($W$) is available and certain conditions on the conditional probabilities of $\mathbb{P}(Y|T, L)$ and $\mathbb{P}(Z, W|T)$ are satisfied. (2) Linear SCMs: They proved that the causal effect of $T$ on $Y$ is identifiable using two proxy variables.

Following their work, Miao et al. (2018) studied a scenario involving two proxy variables, $Z$ and $W$. Unlike the previous results, they allow $Z$ and $W$ to be parent nodes for $T$ and $Y$, respectively. They found that the causal effect can be identified for discrete finite variables if the matrix $\mathbb{P}(W|Z, T = t)$ is invertible. They also provided analogous (nonparametric) conditions for continuous variables. Shi et al. (2020) extended these results, employing a less stringent set of assumptions while still necessitating two proxy variables to identify the causal effect. Later Shuai et al. (2023) considered the setting with one proxy variable and proved that the causal effect is identifiable under the assumption that only the treatment is non-Gaussian, with

the other variables being jointly Gaussian. Cui et al. (2024) proposed an alternative proximal identification procedure to that of Miao et al. (2018), again under the availability of two proxy variables. For lvLiNGAMs, Kivva et al. (2023) gave an explicit moment-based formula for the causal effect when there is no edge from the proxy to the treatment. For a general introduction to proximal causal inference, see also Tchetgen et al. (2024).

Instrumental variables were first introduced in Wright (1928, App. B) and have since become a fundamental identification strategy in both the social sciences (Cunningham, 2021, §7.1) and epidemiology (Didelez & Sheehan, 2007). In linear models, the standard TSLS equations (Angrist & Pischke, 2009, §3.2) have a unique solution only with at least one instrument per treatment. For cases with fewer instruments, Ailer et al. (2023) proposed estimating the causal effect using the minimum norm solution to the TSLS equations, which is always unique but may introduce arbitrary bias. In contrast, Pfister & Peters (2022) showed that, under additional sparsity assumptions, causal effects can be identified by adding an $\ell_0$ penalty to the TSLS equations. For lvLiNGAMs, Silva & Shimizu (2017); Xie et al. (2022) explored the testable implications of instrumental variables.

## 6. Experimental Results[3]

This section presents experimental results on synthetic and experimental data for the graphs studied in Section 3.

As performance metric, we use the relative absolute error

$$\text{err}(\hat{\mathbf{b}}_{Y,T}, \mathbf{b}^*_{Y,T}) := \left| \left( \hat{\mathbf{b}}_{Y,T} - \mathbf{b}^*_{Y,T} \right) / \mathbf{b}^*_{Y,T} \right|,$$

where $\mathbf{b}^*_{Y,T}$ is the true value of causal effect and $\hat{\mathbf{b}}_{Y,T}$ is its estimate. We report the median value of the relative estimation error over 100 random simulations; the filled area on our plots shows the interquartile range of the relative error distribution. Details on the experimental setup and experiments are provided in Appendix D.

### 6.1. Proxy Variable

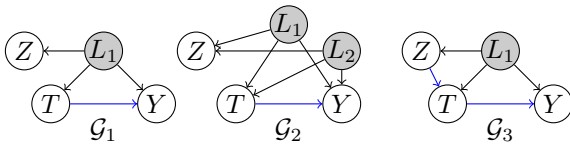

*Figure 5.* The causal graphs considered in the experiments.

We begin with experimental results for the proxy variable settings with the causal graphs illustrated in Fig. 5. We

---

[3]The code to replicate the experiments can be found at
https://github.com/danieletramontano/CEId-from-Moments.

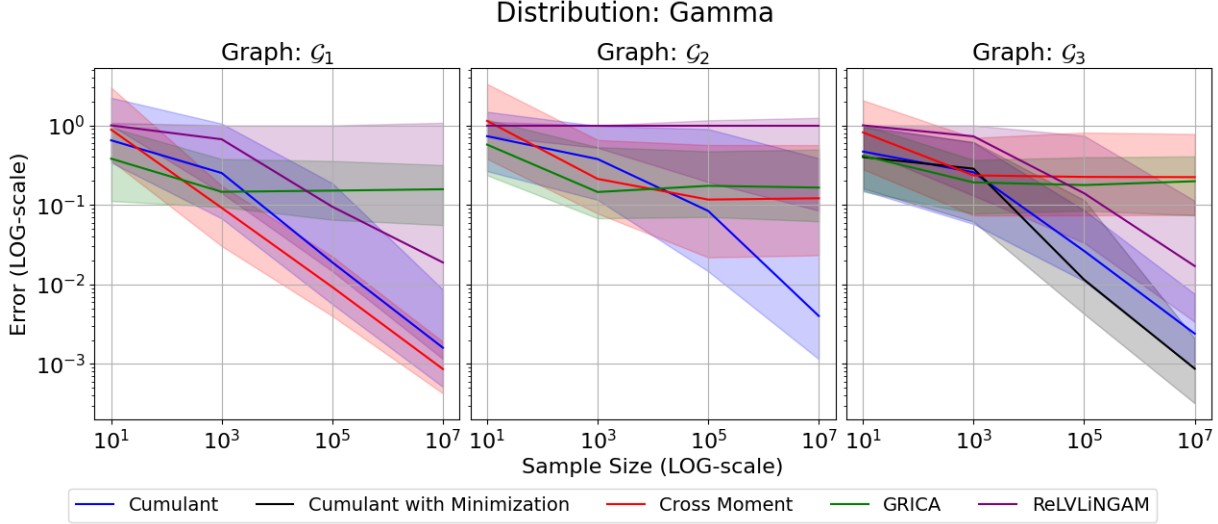

Figure 6. Relative error vs sample size for the graphs in Fig. 5.

compare our method (which we call Cumulant) with the Cross-Moment (Kivva et al., 2023, Alg. 1), GRICA (Tramontano et al., 2024b, §3.5), and ReLVLiNGAM (Schkoda et al., 2024) algorithms.

As can be seen in Fig. 6 (left), for the graph $\mathcal{G}_1$, the Cross-Moment algorithm outperforms all other methods. This is expected since it provides a consistent estimate of the causal effect using third-order cumulants if there is no edge from the proxy variable to the treatment. Although the Cumulant method is also consistent, it uses fourth-order cumulants that are more challenging to estimate.

For the graphs $\mathcal{G}_2$ and $\mathcal{G}_3$, which include either multiple latent variables or a causal edge from $Z$ to $T$, our proposed method significantly outperforms other approaches (see Fig. 6, middle and right). Additionally, an experiment involving both multiple latent variables and a causal edge from $Z$ to $T$ is presented in Fig. 10 in the appendix. For the graph $\mathcal{G}_3$, we also provided the result for the Cumulant method with the minimization technique given in Appendix C.1.1, which improves the performance of the Cumulant method since it reduces the dependency on using the fourth-order cumulants. Notably, for these graphs, neither the Cross-Moment nor the GRICA algorithm provides a consistent estimator of the true causal effect. This can also be seen from the experiments, as the relative error does not decay as the sample size increases. Furthermore, while the ReLVLiNGAM algorithm produces consistent estimators for the causal effect in graphs $\mathcal{G}_1$ and $\mathcal{G}_3$, it performs poorly compared to our method. This results from ReLVLiNGAM performing causal discovery and causal effect estimation simultaneously, increasing its complexity.

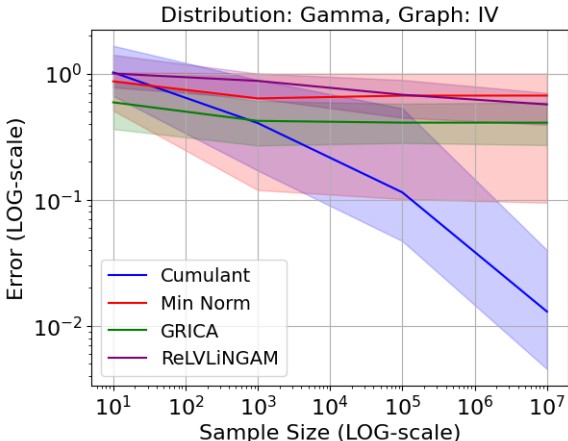

Figure 7. Relative error vs sample size for the graph in Fig. 4.

### 6.2. Underspecified Instrumental Variable

In this part, we provide the experimental results for the underspecified instrumental variable graph depicted in Fig. 4. We compare our method (Cumulant) with the projection on instrument space proposed in Ailer et al. (2023, §3.1) (Min Norm), the GRICA, and the ReLViNGAM algorithm. Fig. 7 shows

$$\left(\text{err}(\hat{\mathbf{b}}_{Y,T^1}, \mathbf{b}^*_{Y,T^1}) + \text{err}(\hat{\mathbf{b}}_{Y,T_2}, \mathbf{b}^*_{Y,T_2})\right)/2$$

against sample size. As can be seen, our method is the only one that consistently estimates the causal effects for the two treatments having access to only one instrument.

*Remark* 6.1 (Small Sample Performance). From Figs. 6 and 7, one can observe that for small sample sizes, the

GRICA method proposed in Tramontano et al. (2024b) exhibits superior performance.

One possible explanation is that cumulant-based methods rely on unbiased estimators of high-order cumulants (typically of order 4 or higher), also known as k-statistics. While these estimators are unbiased, they tend to exhibit high variance when the sample size is small.

In contrast, GRICA solves an optimization problem involving the $\ell_1$-norm of the observed data, which generally has lower sample variance. As a result, GRICA may achieve lower mean-squared error in small-sample regimes due to this variance reduction. However, because the GRICA solution is not asymptotically unbiased, it does not yield a consistent estimator, unlike our proposed method, which retains consistency in the asymptotic limit.

### 6.3. Experiments on Real Data

To assess the practical efficacy of our method, we conduct experiments on the dataset analyzed in Card & Krueger (1993), which contains information on fast-food restaurants in New Jersey and Pennsylvania in 1992. The dataset includes variables such as minimum wage, product prices, store hours, and other relevant features. The original study aimed to estimate the effect of an increase in New Jersey's minimum wage—from \$4.25 to \$5.05 per hour—on employment rates. Importantly, the data were collected both before and after the wage increase in New Jersey, while the minimum wage in Pennsylvania remained constant throughout this period.

For our experiments, we adopt the preprocessing procedure from Kivva et al. (2023). Specifically, we regress the proxy, treatment, and outcome variables on the observed covariates (e.g., product prices, store hours) and then apply our methods on the residuals of these regressions. Assuming that the preprocessed data conform to the causal structures encoded by the graphs $\mathcal{G}_1$ and $\mathcal{G}_2$, we estimate the causal effect to be 2.68 and 2.71, respectively. Prior approaches, such as the cross-moment method (Kivva et al., 2023) and the Difference-in-Differences method, also yield a point estimate of 2.68. In contrast, assuming $\mathcal{G}_3$ as the true graph yields an estimated causal effect of 8.26. Although this still indicates a positive impact of the treatment on the outcome, consistent with prior findings, the magnitude deviates significantly from estimates reported in the literature. A more detailed uncertainty assessment in future work could help clarify the source of this discrepancy.

## 7. Conclusion

We studied causal effect identification and estimation using higher-order cumulants in lvLiNGAM models. We presented novel closed-form solutions for estimating causal

effects in the context of proxy variables and underspecified instrumental variable graphs, which cannot be handled with existing methods. Experimental results demonstrate the accuracy and practical utility of our proposed methods.

## Acknowledgements

This project has received funding from the European Research Council (ERC) under the European Union's Horizon 2020 research and innovation programme (grant agreement No 883818) and supported in part by the SNF project 200021_204355/1, Causal Reasoning Beyond Markov Equivalencies. DT's PhD scholarship is funded by the IGSSE/TUM-GS via a Technical University of Munich–Imperial College London Joint Academy of Doctoral Studies. MD acknowledges support by the DAAD programme Konrad Zuse Schools of Excellence in Artificial Intelligence, sponsored by the Federal Ministry of Education and Research.

## Impact Statement

This paper presents work whose goal is to advance the field of Machine Learning. There are many potential societal consequences of our work, none of which we feel must be specifically highlighted here.

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

# A. Notions of Non-Linear Algebra

In this section, we give the basic definitions of *non-linear* algebra we will need for the proofs; we refer the interested reader to Garcia et al. (2010); Cox et al. (2015); Michałek & Sturmfels (2021) for more details.

**Definition A.1.** For every natural number $n$, we denote the ring of polynomials in $n$ variables $x_1, \ldots, x_n$ by $\mathbb{R}[x_1, \ldots, x_n]$. Let $S$ be a, possibly infinite, subset of $\mathbb{R}[x_1, \ldots, x_n]$. The affine variety associated to it is defined as $\mathcal{V}(S) = \{x \in \mathbb{R}^n \mid f(x) = 0, \forall f \in S\}$. The vanishing ideal associated to a variety $\mathcal{V}$ is $\mathcal{I}(\mathcal{V}) = \{f \in \mathbb{R}[x_1, \ldots, x_n] \mid f(x) = 0 \, \forall x \in \mathcal{V}\}$. The coordinate ring of $\mathcal{V}$ is defined as $\mathbb{R}[\mathcal{V}] = \mathbb{R}[x_1, \ldots, x_n]/\mathcal{I}(\mathcal{V})$.

**Definition A.2.** A term order $\prec$ on the polynomial ring $\mathbb{R}[\mathbf{x}]$ is a total ordering on the monomials in $\mathbb{R}[\mathbf{x}]$ that is compatible with multiplication and such that $1$ is the smallest monomial; that is, $1 = \mathbf{x}^0 \preceq \mathbf{x}^{\mathbf{u}}$ for all $\mathbf{u} \in \mathbb{N}^n$ and if $\mathbf{x}^{\mathbf{u}} \preceq \mathbf{x}^{\mathbf{v}}$, then $\mathbf{x}^{\mathbf{w}} \cdot \mathbf{x}^{\mathbf{u}} \preceq \mathbf{x}^{\mathbf{w}} \cdot \mathbf{x}^{\mathbf{v}}$. Since $\prec$ is a total ordering, every polynomial $g \in \mathbb{R}[\mathbf{x}]$ has a well-defined largest monomial. Let $\mathrm{in}_{\prec}(g)$ be the largest monomial in $g$. For an ideal $I \subseteq \mathbb{R}[\mathbf{x}]$, let $\mathrm{in}_{\prec}(I) = \{\mathrm{in}_{\prec}(g) : g \in I\}$; this is called the *initial ideal* of $I$.

Among the most important term orders is the *lexicographic term order*, which can be defined for any permutation of the variables. In the lexicographic term order, we declare $\mathbf{x}^{\mathbf{u}} \prec \mathbf{x}^{\mathbf{v}}$ if and only if the left-most nonzero entry of $\mathbf{v} - \mathbf{u}$ is positive.

*Elimination orders* are a generalization of the lexicographic order. These are obtained by splitting the variables into a partition $A \cup B$. In the elimination order, $\mathbf{x}^{\mathbf{u}} \prec \mathbf{x}^{\mathbf{v}}$ if $\mathbf{x}^{\mathbf{v}}$ has a larger degree in the $A$ variables than $\mathbf{x}^{\mathbf{u}}$. If $\mathbf{x}^{\mathbf{v}}$ and $\mathbf{x}^{\mathbf{u}}$ have the same degree in the $A$ variables, then some other term order is used to break ties.

**Definition A.3.** A finite subset $G \subseteq I$ is called a *Gröbner basis* for $I$ with respect to the term order $\prec$ if

$$\mathrm{in}_{\prec}(I) = \{\mathrm{in}_{\prec}(g) : g \in G\}.$$

The Gröbner basis is called *reduced* if the coefficient of $\mathrm{in}_{\prec}(g)$ in $g$ is 1 for all $g$, each $\mathrm{in}_{\prec}(g)$ is a minimal generator of $\mathrm{in}_{\prec}(I)$, and no terms besides the initial terms of $G$ belong to $\mathrm{in}_{\prec}(I)$.

**Lemma A.4** (Okamoto, 1973, Lemma). *Let $f(x_1, \ldots, x_n)$ be a polynomial in real variables $x_1, \ldots, x_n$, which is not identically zero. The set of zeros of the polynomial is a Lebesgue measure zero subset of $\mathbb{R}^n$.*

**Lemma A.5.** *Let $\mathbb{R}_{\mathbf{A}}^{\mathcal{G}}$ and $\mathbb{R}^{\mathcal{G}}$ defines as in Section 2.2. Then we have $\mathbb{R}^{\mathcal{G}} \sim \mathbb{R}_{\mathbf{A}}^{\mathcal{G}} \sim \mathbb{R}^{|e|}$, where with the symbol $\sim$, we denote an isomorphism of affine varieties, see, e.g., Cox et al. (2015, Def. 6, §5) for a definition. Moreover $\mathbb{R}[\mathcal{G}]$, $\mathbb{R}[\mathcal{G}_{\mathbf{A}}]$, and $\mathbb{R}[a_{i,j} \mid j \to i \in \mathcal{G}]$ are isomorphic as rings.*

*Proof.* The isomorphism $\mathbb{R}_{\mathbf{A}}^{\mathcal{G}} \sim \mathbb{R}^{|e|}$ comes directly from its definition. Indeed it is easy to see that $\mathbb{R}_{\mathbf{A}}^{\mathcal{G}}$ is an $|e|$-dimensional linear subspace of $\mathbb{R}^{p \times p} = (a_{i,j})_{i,j \in p \times p}$, defined by the linear equations $a_{i,i} = 1$, and $a_{i,j} = 0$, $\forall i, j \in \mathcal{V}$ such that $j \to i \notin \mathcal{G}$.

To prove the isomorphism $\mathbb{R}^{\mathcal{G}} \sim \mathbb{R}_{\mathbf{A}}^{\mathcal{G}}$, we need to prove that there is a polynomial bijective map between the two spaces. From (4), and using $[\mathbf{B}_o]_{i,j} = [(\mathbf{A}_{o,o})^{-1}]_{i,j} = (-1)^{i+j} \det([\mathbf{A}_{o,o}]_{\backslash j, \backslash i})$, where we used that $\det(\mathbf{A}_{o,o}) = 1$. It is clear that $\mathbb{R}^{\mathcal{G}}$ is the image of polynomial map of $\mathbb{R}_{\mathbf{A}}^{\mathcal{G}}$. Let us call this polynomial map $\psi$ and assume $\psi(\mathbf{A}) = \psi(\tilde{\mathbf{A}})$. Then from the definition of $\psi$ we have $(I - \mathbf{A}_{o,o})^{-1} = (I - \tilde{\mathbf{A}}_{o,o})^{-1}$ that implies $\mathbf{A}_{o,o} = \tilde{\mathbf{A}}_{o,o}$. Moreover, $(I - \mathbf{A}_{o,o})^{-1}\mathbf{A}_{o,l} = (I - \tilde{\mathbf{A}}_{o,o})^{-1}\tilde{\mathbf{A}}_{o,l}$ that implies $\mathbf{A}_{o,l} = \tilde{\mathbf{A}}_{o,l}$ and so $\mathbf{A} = \tilde{\mathbf{A}}$.

The isomorphisms between the rings come from Cox et al. (2015, §5, Thm. 9). $\qquad\square$

**Corollary A.6.** *Let $f \in \mathbb{R}[\mathcal{G}]$ be a non-zero polynomial. Then the subset of $\mathbb{R}^{\mathcal{G}}$ on which $f$ vanishes is a Lebesgue measure $0$ subset of $\mathbb{R}^{\mathcal{G}}$.*

*Proof.* Thanks to the isomorphism in Lemma A.5, we can apply Lemma A.4 to $\mathbb{R}^{\mathcal{G}}$. $\qquad\square$

**Definition A.7.** Let $\pi \in \mathcal{P}(j, i)$. The path monomial associated to it is defined as

$$a^{\pi} = a_{i_1, i_2} \cdot \cdots \cdot a_{i_k, i_{k+1}} \in \mathbb{R}[\mathcal{G}_{\mathbf{A}}].$$

**Lemma A.8.** *Let $\mathbf{A}$ defined as in (2). We have*

$$\mathbf{B} = (I - \mathbf{A})^{-1} = \sum_{i=0}^{\infty} \mathbf{A}^i = I + \mathbf{A} + \mathbf{A}^2 + \cdots + \mathbf{A}^p,$$

$$\mathbf{b}_{i,j} = \sum_{P \in \mathcal{P}(i,j)} a^P.$$

*In particular* $\mathbf{b}_{i,j} = 0 \in \mathbb{R}[\mathcal{G}_\mathbf{A}]$ *if and only if* $P \in \mathcal{P}(i,j) = \emptyset$.

## B. Additional Proofs

*Remark* B.1. The polynomial $p_{\mathbf{V}_o,l}(\mathbf{b})$ mentioned in Theorem 3.1, can be obtained as the determinant of an $l+2 \times l+2$ minor the following matrix containing the first row

$$
\begin{bmatrix}
1 & \mathbf{b} & \cdots & \mathbf{b}^{l+2} \\
\hline
\mathbf{c}^{l+2}(\mathbf{V}_o)_{1,\ldots,1} & \mathbf{c}^{l+2}(\mathbf{V}_o)_{1,\ldots,1,2} & \cdots & \mathbf{c}^{l+2}(\mathbf{V}_o)_{1,2,\ldots,2} \\
\hline
\mathbf{c}^{l+3}(\mathbf{V}_o)_{1,1,\ldots,1} & \mathbf{c}^{l+3}(\mathbf{V}_o)_{1,1,\ldots,1,2} & \cdots & \mathbf{c}^{l+3}(\mathbf{V}_o)_{1,1,2,\ldots,2} \\
\mathbf{c}^{l+3}(\mathbf{V}_o)_{2,1,\ldots,1} & \mathbf{c}^{l+3}(\mathbf{V}_o)_{2,1,\ldots,1,2} & \cdots & \mathbf{c}^{l+3}(\mathbf{V}_o)_{2,1,2,\ldots,2} \\
\hline
\vdots & \vdots & \ddots & \vdots \\
\hline
\mathbf{c}^{k(l)}(\mathbf{V}_o)_{1,\ldots,1,1,1,\ldots,1,1} & \mathbf{c}^{k(l)}(\mathbf{V}_o)_{1,\ldots,1,1,1,\ldots,1,2} & \cdots & \mathbf{c}^{k(l)}(\mathbf{V}_o)_{1,\ldots,1,1,2,\ldots,2,2} \\
\vdots & \vdots & \ddots & \vdots \\
\mathbf{c}^{k(l)}(\mathbf{V}_o)_{2,\ldots,2,1,1,\ldots,1,1} & \mathbf{c}^{k(l)}(\mathbf{V}_o)_{2,\ldots,2,1,1,\ldots,1,2} & \cdots & \mathbf{c}^{k(l)}(\mathbf{V}_o)_{2,\ldots,2,1,2,\ldots,2,2}
\end{bmatrix}.
$$

The proof of this fact can be found in Schkoda et al. (2024, Thm. 4).

**Lemma B.2.** *Let* $\mathbf{V}_o = [Z,T,Y]$ *be a vector generated from a lvLiNGAM model compatible with the graph in Fig. 3, and let* $\mathbf{b}$ *be either equal to* $[\mathbf{b}_{T,Z}, \mathbf{b}_{Y,Z}]$ *or to* $[\mathbf{b}_{T,L_i}, \mathbf{b}_{Y,L_i}]$ *for some* $i \in [l]$. *Then, the triple*

$$
\mathbf{V}^\mathbf{b} := [Z, T - \mathbf{b}_1 Z, Y - \mathbf{b}_2 Z]
$$

*follows a lvLiNGAM model compatible with the graph in Fig. 2 with the causal effect from* $T - \mathbf{b}_1 Z$ *to* $Y - \mathbf{b}_2 Z$ *being the same as in the original model.*

*Proof.* From (3), we know that

$$
\mathbf{V}_o = \begin{bmatrix}
1 & 1 & \cdots & 1 & 0 & 0 \\
\mathbf{b}_{T,Z} & \mathbf{b}_{T,L_1} & \cdots & \mathbf{b}_{T,L_l} & 1 & 0 \\
\mathbf{b}_{Y,Z} & \mathbf{b}_{Y,L_1} & \cdots & \mathbf{b}_{Y,L_l} & \mathbf{b}_{Y,T} & 1
\end{bmatrix}
\begin{bmatrix}
\mathbf{N}_Z \\ \mathbf{N}_{L_1} \\ \vdots \\ \mathbf{N}_{L_l} \\ \mathbf{N}_T \\ \mathbf{N}_Y
\end{bmatrix}.
$$

From simple linear algebra manipulation, it follows that

$$
\mathbf{V}^\mathbf{b} = \begin{bmatrix} 1 & 0 & 0 \\ -\mathbf{b}_1 & 1 & 0 \\ -\mathbf{b}_2 & 0 & 1 \end{bmatrix} \mathbf{V}_o = \begin{bmatrix}
1 & 1 & \cdots & 1 & 0 & 0 \\
-\mathbf{b}_1 + \mathbf{b}_{T,Z} & -\mathbf{b}_1 + \mathbf{b}_{T,L_1} & \cdots & -\mathbf{b}_1 + \mathbf{b}_{T,L_l} & 1 & 0 \\
-\mathbf{b}_2 + \mathbf{b}_{Y,Z} & -\mathbf{b}_2 + \mathbf{b}_{Y,L_1} & \cdots & -\mathbf{b}_2 + \mathbf{b}_{Y,L_l} & \mathbf{b}_{Y,T} & 1
\end{bmatrix}
\begin{bmatrix}
\mathbf{N}_Z \\ \mathbf{N}_{L_1} \\ \vdots \\ \mathbf{N}_{L_l} \\ \mathbf{N}_T \\ \mathbf{N}_Y
\end{bmatrix}.
$$

By setting $\mathbf{b}$ to be either equal to $[\mathbf{b}_{T,Z}, \mathbf{b}_{Y,Z}]$ or to $[\mathbf{b}_{T,L_i}, \mathbf{b}_{Y,L_i}]$, we set one of the first $l+1$ columns of the mixing matrix corresponding to $\mathbf{V}^\mathbf{b}$ to $[1,0,0]$, hence removing the edge from $Z$ to $T$. $\square$

**Lemma B.3.** *Let* $\mathbf{V}_o = [Z, T, Y]$ *be a vector generated from a lvLiNGAM model compatible with the graph in Fig. 3 with one latent variable, and let* $q_{c^2(\mathbf{V}_o)}(\mathbf{b})$ *be the following univariate polynomial*

$$q_{c^2(\mathbf{v}_o)}(\mathbf{b}) := \frac{\mathbf{c}^2(\mathbf{V}_o)_{T,Y} - \mathbf{b} \cdot \mathbf{c}^2(\mathbf{V}_o)_{Z,Y}}{\mathbf{c}^2(\mathbf{V}_o)_{T,T} - \mathbf{b} \cdot \mathbf{c}^2(\mathbf{V}_o)_{Z,T}}. \tag{15}$$

*Then, we have* $q_{c^2(\mathbf{V}_o)}(\mathbf{b}_{T,L_1}) = \mathbf{b}_{Y,T}$.

*Proof.* Direct computation, applying Lemma 2.2 and Lemma A.8. □

**Lemma B.4.** *Let* $\mathbf{V}_o = [I, T^1, \ldots, T^k, Y]$ *be a vector generated from a lvLiNGAM model compatible with an instrumental variable graph. Consider now the variables*

$$T^{I,i} = T^i - \mathbf{b}_{T^i,I} I, \quad Y^I = Y - \mathbf{b}_{Y,I} I,$$

*obtained by regressing out* $I$ *from* $T^i$ *and* $Y$*, respectively.*

*Each one of the pairs* $[T^{I,i}, Y^I]$ *can be represented by a lvLiNGAM model with two observed variables and at most l latent confounders, with the causal effect from* $T^{I,i}$ *to* $Y^I$ *being the same as in the original distribution.*

*Proof.* From (3), we know that

$$\begin{bmatrix} I \\ T^i \\ Y \end{bmatrix} = \begin{bmatrix} 1 & 0 & \cdots & 0 & 0 & 0 \\ \mathbf{b}_{T,I} & \mathbf{b}_{T,L_1} & \cdots & \mathbf{b}_{T,L_l} & 1 & 0 \\ \mathbf{b}_{Y,I} & \mathbf{b}_{Y,L_1} & \cdots & \mathbf{b}_{Y,L_l} & \mathbf{b}_{Y,T} & 1 \end{bmatrix} \begin{bmatrix} \mathbf{N}_I \\ \mathbf{N}_{L_1} \\ \vdots \\ \mathbf{N}_{L_l} \\ \mathbf{N}_T \\ \mathbf{N}_Y \end{bmatrix}.$$

From simple linear algebra manipulation, it follows that

$$\begin{bmatrix} T^{I,i} \\ Y^I \end{bmatrix} = \begin{bmatrix} -\mathbf{b}_{T^i,I} & 1 & 0 \\ -\mathbf{b}_{Y,I} & 0 & 1 \end{bmatrix} \begin{bmatrix} I \\ T^i \\ Y \end{bmatrix} = \begin{bmatrix} 0 & -\mathbf{b}_{T^i,I} + \mathbf{b}_{T,L_1} & \cdots & -\mathbf{b}_{T^i,I} + \mathbf{b}_{T,L_l} & 1 & 0 \\ 0 & -\mathbf{b}_{Y,I} + \mathbf{b}_{Y,L_1} & \cdots & -\mathbf{b}_{Y,I} + \mathbf{b}_{Y,L_l} & \mathbf{b}_{Y,T} & 1 \end{bmatrix} \begin{bmatrix} \mathbf{N}_I \\ \mathbf{N}_{L_1} \\ \vdots \\ \mathbf{N}_{L_l} \\ \mathbf{N}_T \\ \mathbf{N}_Y \end{bmatrix}$$

$$= \begin{bmatrix} -\mathbf{b}_{T^i,I} + \mathbf{b}_{T,L_1} & \cdots & -\mathbf{b}_{T^i,I} + \mathbf{b}_{T,L_l} & 1 & 0 \\ -\mathbf{b}_{Y,I} + \mathbf{b}_{Y,L_1} & \cdots & -\mathbf{b}_{Y,I} + \mathbf{b}_{Y,L_l} & \mathbf{b}_{Y,T} & 1 \end{bmatrix} \begin{bmatrix} \mathbf{N}_{L_1} \\ \vdots \\ \mathbf{N}_{L_l} \\ \mathbf{N}_T \\ \mathbf{N}_Y \end{bmatrix}.$$

Which is indeed compatible with the graph in Fig. 1. □

**Theorem.** *Let* $\mathcal{G}_{IV}$ *be an instrumental variable graph, with instrument* $I$*, treatments* $T^1, \ldots, T^k$*, and outcome* $Y$*, and let* $l := \max_{i \in [n]} |\operatorname{an}(T^i) \cap \operatorname{an}(Y) \setminus I|$*. Then, the causal effect from* $T^i$ *to* $Y$ *is generically identifiable from the first* $k(l)$ *cumulants of the distribution.*

*Proof of Theorem 3.7.* Since $\operatorname{an}(I) \cap \operatorname{an}(T^i) \cap \mathcal{L} = \operatorname{an}(I) \cap \operatorname{an}(Y) \cap \mathcal{L} = \emptyset$, we can identify $\mathbf{b}_{T^i,I}$ and $\mathbf{b}_{Y,I}$ from the covariance matrix through backdoor adjustment (Pearl et al. (2016, §3.3), Henckel et al. (2022, Prop. 1)). From Ailer et al. (2023, § 3.1), we know that the causal effects of interest satisfy the following equation:

$$r_I(\mathbf{b}) := \mathbf{b}_{Y,I} - \sum_i \mathbf{b}_{T^i,I} \mathbf{b}_{Y,T^i} = 0 \in \mathbb{R}[\mathcal{G}], \tag{16}$$

where $\mathbf{b} = [\mathbf{b}_{Y,T^1}, \ldots, \mathbf{b}_{Y,T^k}]$. Consider now the variables

$$T^{I,i} = T^i - \mathbf{b}_{T^i,I}I, \quad Y^I = Y - \mathbf{b}_{Y,I}I, \tag{17}$$

obtained by regressing out $I$ from $T^i$ and $Y$, respectively.

Each one of the pairs $[T^{I,i}, Y^I]$ can be represented by a lvLiNGAM model with two observed variables and at most $l$ latent confounders, with the causal effect from $T^{I,i}$ to $Y^I$ being the same as in the original distribution (Lemma B.4).

Using Theorem 3.1, we know that the vector

$$\mathbf{b}^i := [\mathbf{b}_{Y,T^i}, \mathbf{b}_{Y,L_1}, \ldots, \mathbf{b}_{Y,L_l}] \tag{18}$$

can be obtained as roots of a degree $l+1$ polynomial constructed using cumulants up to order $k(l)$ of the observational distribution (up to some permutations).

Consider the polynomial $r_I(b_1, \ldots, b_n) \in \mathbb{R}[b_1, \ldots, b_n]$ defined in (16). For every choice of $\mathbf{b} \in \mathbf{b}^1 \times \cdots \times \mathbf{b}^n$, $r_I(\mathbf{b})$ defines a different polynomial in $\mathbb{R}[\mathcal{G}]$. We have already seen, that for $\mathbf{b} = [\mathbf{b}_{Y,T^1}, \ldots, \mathbf{b}_{Y,T^k}]$ this defines the zero polynomial. To conclude, it is only left to show that

$$r_I(\mathbf{b}) \neq 0 \in \mathbb{R}[\mathcal{G}] \qquad \forall \mathbf{b} \in \mathbf{b}^1 \times \cdots \times \mathbf{b}^n \setminus \{[\mathbf{b}_{Y,T^1}, \ldots, \mathbf{b}_{Y,T^k}]\}, \tag{19}$$

the result will follow by applying Lemma A.4.

Let us rewrite the entries of $\mathbf{b}^i$ as $\mathbf{b}_{Y,T^i} + c_i(\mathbf{b}_{Y,L_i} - \mathbf{b}_{Y,T^i})$ for some $c_i \in \{0,1\}$. This way, we can write $r_I(\mathbf{b})$ as

$$\mathbf{b}_{I,Y} - \sum_i \mathbf{b}_{I,T^i}(\mathbf{b}_{Y,T^i} + c_{j_i}(\mathbf{b}_{Y,L_{j_i}} - \mathbf{b}_{Y,T^i})) = -\sum_i c_{j_i}\mathbf{b}_{I,T^i}(\mathbf{b}_{Y,L_{j_i}} - \mathbf{b}_{Y,T^i}),$$

using Lemmas A.5 and A.8 we can rewrite it as

$$\sum_i c_{j_i} \left( \sum_{\pi_i \in \mathcal{P}(I,T^i)} \mathbf{a}^{\pi_i} \right) \left( \sum_{\pi_{j_i} \in \mathcal{P}(Y,L_{j_i})} \mathbf{a}^{\pi_{j_i}} - \sum_{\pi_{Y,i} \in \mathcal{P}(T^i,Y)} \mathbf{a}^{\pi_{Y,T^i}} \right) \in \mathbb{R}[\mathcal{G}_\mathbf{A}].$$

Notice that every summand in the above equation is a monomial of degree at least two. If $c_{j_i} = 1$ for some $i$, then the degree two term $\mathbf{a}_{I,T^i}\mathbf{a}_{T^i,Y}$ appears only once as a summand. This implies that the last equation defines a non-zero polynomial in $\mathbb{R}_\mathbf{A}^\mathcal{G}$, which concludes the proof. $\square$

*Remark* B.5 (Multiple Instruments). For simplicity, we stated and proved the theorem for the case of a single instrumental variable. However, the result naturally extends to scenarios with multiple valid instruments $I$, provided that each treatment has at least one valid instrument.

To adapt the proof, (19) should be replaced with

$$T^{I,i} = T^i - \sum_{j \in \mathcal{I}_i} \mathbf{b}_{T^i,I^j}I^j, \quad Y^I = Y - \sum_{j \in [s]} \mathbf{b}_{Y,I^j}I^j, \tag{20}$$

where $\mathcal{I}_i$ is the set of valid instruments for the treatment $T^i$.

Additionally, the variety $\mathcal{V}_I := \mathcal{V}(r_{I^1}(\mathbf{b}), \ldots, r_{I^s}(\mathbf{b}))$ should be used in place of the single polynomial $r_I(\mathbf{b})$ in (19).

## C. Estimation

### C.1. Proxy Variable with an Edge from Proxy to Treatment

Algorithm 2 outlines the estimation procedure for causal effect estimation corresponding to the graph in Fig. 3. This algorithm replaces the steps in the proof of Theorem 3.5 with their respective finite-sample versions.

Specifically, lines 1 and 3 correspond to (13). Lines 3 to 5 align with (14), where the minimization step in line 5 is equivalent to that in line 8 of Algorithm 1 and is further described in Section 4. The for loop spanning lines 7 to 17 corresponds to applying Algorithm 1 for all possible choices of $\mathbf{b}$ in (12).

---

**Algorithm 2** Proxy Variable with an Edge from Proxy to Treatment (Fig. 3)

---

**INPUT:** Data $\mathbf{V}_n = [Z_n, T_n, Y_n]$, bound on the number of latent variables $l$.

1:   $\mathbf{b}_n^T \leftarrow$ roots of $p_{[Z_n,T_n],l}(\mathbf{b}) = 0$ $\{(13)\}$
2:   $\mathbf{b}_n^Y \leftarrow$ roots of $p_{[Z_n,Y_n],l}(\mathbf{b}) = 0$ $\{(13)\}$
3:   $\mathbf{c}_{T_n}^{l+1} \leftarrow$ solution to the linear system $\mathrm{M}(\mathbf{b}_n^{ZT}, l+1) \cdot \mathbf{c}^{l+1} = \mathbf{c}_{(1,2)}^{l+1}([Z_n, T_n])$ $\{(8)\}$
4:   $\mathbf{c}_{Y_n}^{l+1} \leftarrow$ solution to the linear system $\mathrm{M}(\mathbf{b}_n^{ZY}, l+1) \cdot \mathbf{c}^{l+1} = \mathbf{c}_{(1,2)}^{l+1}([Z_n, Y_n])$ $\{(8)\}$
5:   $\sigma_n \leftarrow \arg\min_{\sigma \in S_{l+1}}(\|\mathbf{c}_{T_n}^{l+1} - \sigma(\mathbf{c}_{Y_n}^{l+1})\|_2^2)$
6:   $r_{\min} \leftarrow \infty$
7:   **for all** $i$ in $[l+1]$ **do**
8:      $\hat{\mathbf{b}}_{T,Z} \leftarrow \mathbf{b}_n^{ZT}[i]$
9:      $\hat{\mathbf{b}}_{Y,Z} \leftarrow \mathbf{b}_n^{ZY}[\sigma(i)]$
10:     $\hat{\mathbf{V}}_n \leftarrow [Z_n, T_n - \hat{\mathbf{b}}_{T,Z} Z_n, Y_n - \hat{\mathbf{b}}_{Y,Z} Z_n]$ $\{(12)\}$
11:     $\hat{\mathbf{b}}_{Y,T} \leftarrow Algorithm\ 1(\hat{\mathbf{V}}_n, l)$ {Lemma B.2}
12:     $r \leftarrow |\hat{\mathbf{b}}_{Y,Z} - \hat{\mathbf{b}}_{Y,T} * \hat{\mathbf{b}}_{T,Z}|$
13:     **if** $r < r_{\min}$ **then**
14:        $r_{\min} \leftarrow r$
15:        $\mathbf{b}_{Y,T}^n \leftarrow \hat{\mathbf{b}}_{Y,T}$
16:     **end if**
17: **end for**
18: **RETURN:** $\mathbf{b}_{Y,T}^n$

---

At the population level, any choice of $\mathbf{b}$ results in the correct causal effect. However, in practice, we observed that using the sample version of $[\mathbf{b}_{T,Z}, \mathbf{b}_{Y,Z}]$ yields better performance. Among the pairs in (14), $[\mathbf{b}_{T,Z}, \mathbf{b}_{Y,Z}]$ is the only one satisfying the equation $\mathbf{b}[2] - \mathbf{b}_{Y,T} \cdot \mathbf{b}[1] = 0$ (Lemma A.8). Therefore, we select the estimate derived from the pair that minimizes the sample version of this equation. This explains the steps outlined in lines 13 to 18.

### C.1.1. PROXY VARIABLE WITH AN EDGE FROM PROXY TO TREATMENT WITH ONE LATENT VARIABLE

In this section we present two specialized estimation procedures for the causal effect in Fig. 2 with one latent variable.

First, Algorithm 3 is a simplified version of Algorithm 2, tailored for the case with a single latent confounder. The key distinction between the two procedures lies in how the candidate value for the causal effect is computed: Algorithm 3 utilizes Lemma B.3 (lines 10–11 of Algorithm 3), whereas Algorithm 2 relies on Lemma B.2 (line 11 of Algorithm 2).

Next, we introduce an optimization technique that leverages cumulants up to degree three. While Theorem 3.6 establishes that the causal effect is not identifiable using second- and third-order cumulants alone, we observe that this procedure often achieves better finite-sample performance when initialized with a reliable starting point, compared to directly applying Algorithm 3.

Let $\mathbf{V}o = [Z, T, Y]$ be a vector generated from a lvLiNGAM model compatible with the graph in Fig. 3 with one latent variable. The following objective function is used:

$$h_{\mathbf{V}_o, \hat{\mathbf{b}}_{YT}}(\mathbf{b}) := \left(\mathbf{b} - \frac{\mathbf{c}^2(\mathbf{V}_o)_{T,Y} - g(\mathbf{b})\mathbf{c}^2(\mathbf{V}_o)_{Z,Y}}{\mathbf{c}^2(\mathbf{V}_o)_{T,T} - g(\mathbf{b})\mathbf{c}^2(\mathbf{V}_o)_{Z,T}}\right)^2 + \left(\mathbf{b} - \hat{\mathbf{b}}_{YT}\right)^2, \tag{21}$$

where

$$g(\mathbf{b}) = \frac{\mathbf{c}_{1,3,3}^{(3)}(\mathbf{V}_o^{\mathbf{b}})\mathbf{c}_{2,2,3}^{(3)}(\mathbf{V}_o^{\mathbf{b}})}{\mathbf{c}_{1,1,3}^{(3)}(\mathbf{V}_o^{\mathbf{b}})\mathbf{c}_{2,3,3}^{(3)}(\mathbf{V}_o^{\mathbf{b}})}, \qquad \mathbf{V}_o^{\mathbf{b}} := [Z, T, Y - \mathbf{b}T]$$

Using Lemma 2.3, it can be shown that, if $\mathbf{c}^{(3)}(L_1)_{1,1,1} \neq 0$, then $g(\mathbf{b}_{Y,T}) = \mathbf{b}_{T,L_1}$. As a result, Lemma B.3 guarantees that $\mathbf{b}_{Y,T}$ minimizes the first term in (21). The second term in (21) serves as a regularization term to ensure the solution remains close to the initial estimate.

---

**Algorithm 3** Proxy Variable with an Edge from Proxy to Treatment with one Latent (Fig. 3)

---

**INPUT:** Data $\mathbf{V}_n = [Z_n, T_n, Y_n]$.

1: $\mathbf{b}_n^T \leftarrow$ roots of $p_{[Z_n, T_n], 1}(\mathbf{b}) = 0$ {(13)}
2: $\mathbf{b}_n^Y \leftarrow$ roots of $p_{[Z_n, Y_n], 1}(\mathbf{b}) = 0$ {(13)}
3: $\mathbf{c}_{T_n}^2 \leftarrow$ solution to the linear system $\mathrm{M}(\mathbf{b}_n^{ZT}, 2) \cdot \mathbf{c}^2 = \mathbf{c}_{(1,2)}^2([Z_n, T_n])$ {(8)}
4: $\mathbf{c}_{Y_n}^2 \leftarrow$ solution to the linear system $\mathrm{M}(\mathbf{b}_n^{ZY}, 2) \cdot \mathbf{c}^2 = \mathbf{c}_{(1,2)}^2([Z_n, Y_n])$ {(8)}
5: $\sigma_n \leftarrow \arg\min_{\sigma \in S_2}(||\mathbf{c}_{T_n}^2 - \sigma(\mathbf{c}_{Y_n}^2)||_2^2)$
6: $\mathbf{b}_{T,Z}^1 \leftarrow \mathbf{b}_n^{ZT}[1]$
7: $\mathbf{b}_{Y,Z}^1 \leftarrow \mathbf{b}_n^{ZY}[\sigma(1)]$
8: $\mathbf{b}_{T,Z}^2 \leftarrow \mathbf{b}_n^{ZT}[2]$
9: $\mathbf{b}_{Y,Z}^2 \leftarrow \mathbf{b}_n^{ZY}[\sigma(2)]$
10: $\mathbf{b}_{Y,T}^1 \leftarrow q_{c^2(\mathbf{V}_n)}(\mathbf{b}_{T,Z}^2)$ {(15), Lemma B.3}
11: $\mathbf{b}_{Y,T}^2 \leftarrow q_{c^2(\mathbf{V}_n)}(\mathbf{b}_{T,Z}^1)$ {(15), Lemma B.3}
12: $r^1 \leftarrow |\mathbf{b}_{Y,Z}^1 - \mathbf{b}_{Y,T}^1 \cdot \mathbf{b}_{T,Z}^1|$
13: $r^2 \leftarrow |\mathbf{b}_{Y,Z}^2 - \mathbf{b}_{Y,T}^2 \cdot \mathbf{b}_{T,Z}^2|$
14: **if** $r^1 < r^2$ **then**
15:     **RETURN:** $\mathbf{b}_{Y,T}^1$
16: **end if**
17: **RETURN:** $\mathbf{b}_{Y,T}^2$

---

**Algorithm 4** Proxy Variable with an Edge from Proxy to Treatment with one Latent with Optimization (Fig. 3)

---

**INPUT:** Data $\mathbf{V}_n = [Z_n, T_n, Y_n]$.

1: $\hat{\mathbf{b}}_{Y,T} \leftarrow Algorithm\ 3(\mathbf{V}_n)$
2: $\mathbf{b}_{Y,T}^n \leftarrow \arg\min_{\mathbf{b} \in \mathbb{R}} h_{\mathbf{V}_n, \hat{\mathbf{b}}_{YT}}(\mathbf{b})${(21)}
3: **RETURN:** $\mathbf{b}_{Y,T}^n$

---

In practice, we solve the optimization problem using the Python implementation of the BFGS algorithm (Nocedal & Wright, 2006, §6.1) provided in Jones et al. (2001–). The finite-sample version of this optimization process is detailed in Algorithm 4. *Remark* C.1. If $\mathbf{c}^{(3)}(L_1)_{1,1,1}$ is zero, higher-order cumulants can be used to construct $g(\mathbf{b})$. The existence of such a polynomial is guaranteed as long as $L_1$ is non-Gaussian; see, for example, Kivva et al. (2023, Thm. 1).

### C.2. Underspecified Instrumental Variable

Algorithm 5 outlines the estimation procedure for causal effect estimation corresponding to the graph in Fig. 8 with one instrumental variable. This algorithm replaces the steps in the proof of Theorem 3.7 with their respective finite-sample versions.

Specifically, lines 1 to 9 involve computing the covariance matrix and performing the regression adjustments required to derive the finite-sample versions of the vectors described in (17).

The for loop in lines 11 to 17 evaluates the finite-sample approximation of the polynomial $r_I(\mathbf{b})$ defined in (16). As the estimate of the causal effect, the algorithm selects the projection over the line defined by the equation $r_I(\mathbf{b}) = 0$ of the tuple $\mathbf{b}_n$ that minimizes $r_I(\mathbf{b}_n)$.

Algorithm 6 is an extension of Algorithm 5 that accommodates the presence of multiple instruments together. It implements adaptations described in Remark B.5.

## D. Details on the Experimental Setting and Additional Experiments

All the experiments in this subsection are done on the synthetic data generated according to the specific causal structure established for it. To generate synthetic data, we specify all exogenous noises from the same family of distributions (with parameters sampled according to Table 1) and select all non-zero entries within the matrix $\mathbf{A}$ through uniform sampling

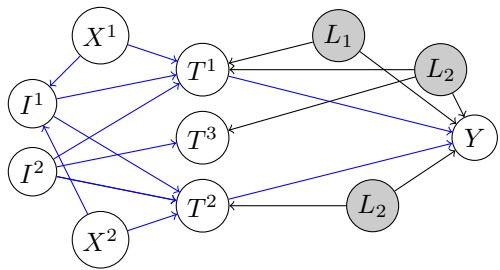

*Figure 8.* An example of a causal graph for the underspecified instrumental variable model.

---

**Algorithm 5** Underspecified Instrumental Variables (Fig. 8)

---

**INPUT:** Data $\mathbf{V}_n = [I_n, T_n^1 \ldots, T_n^k, Y_n, X^1, \ldots, X^e]$, the causal graph $\mathcal{G}$, bound on the number of latent variables $l$.

1: $\Sigma_n \leftarrow \mathbf{c}^{(2)}(\mathbf{V}_n)$ {Sample covariance matrix}
2: $\mathrm{ad}(I, Y) \leftarrow \mathrm{an}(I) \cap \mathrm{an}(Y) \cap \mathcal{O}$ {Valid adjustment set}
3: $\mathbf{b}_{Y,I,n} \leftarrow (\Sigma_n)_{Y,I|\mathrm{ad}(I,Y)}/(\Sigma_n)_{I,I|\mathrm{ad}(I,Y)}$ {Regression adjustment (Henckel et al., 2022, Prop. 1)}
4: $Y_n^I \leftarrow Y_n - \mathbf{b}_{Y,I,n}I_n$ {(17)}
5: **for all** $i \in [k]$ **do**
6:    $\mathrm{ad}(I, T^i) \leftarrow \mathrm{an}(I) \cap \mathrm{an}(T^i) \cap \mathcal{O}$
7:    $\mathbf{b}_{T^i,I,n} \leftarrow (\Sigma_n)_{T^i,I|\mathrm{ad}(I,T^i)}/(\Sigma_n)_{I,I|\mathrm{ad}(I,Y)}$
8:    $T_n^{I,i} \leftarrow T_n^i - \mathbf{b}_{T^i,I,n}I_n$
9:    $\mathbf{b}_n^i \leftarrow \text{roots of } p_{[T_n^{I,i}, Y_n^I], l}(\mathbf{b}) = 0$ {(18)}
10: **end for**
11: $r_{\min} \leftarrow \infty$
12: **for all** $\mathbf{b}_n \in \mathbf{b}_n^1 \times \cdots \times \mathbf{b}_n^k$ **do**
13:    $r_b \leftarrow |r_I(\mathbf{b}_n)|$ {(19)}
14:    **if** $r_b < r_{\min}$ **then**
15:       $r_{\min} \leftarrow r_b$
16:       $\mathbf{b}_{Y,T^1,\ldots,T^k}^n \leftarrow \underset{\mathbf{b}\,:\,r_I(\mathbf{b})=0}{\arg\min} ||\mathbf{b} - \mathbf{b}_n||_2^2$
17:    **end if**
18: **end for**
19: **RETURN:** $\mathbf{b}_{Y,T^1,\ldots,T^k}^n$

---

from $[-0.9, -0.5] \cup [0.5, 0.9]$.

In the figures, we plot the median relative error over 100 independent experiments; the filled area on our plots shows the interquartile range.

**Algorithm 6** Underspecified Instrumental Variables with Multiple Instruments (Fig. 8)

**INPUT:** Data $\mathbf{V}_n = [I_n^1, \ldots, I_n^s, T_n^1 \ldots, T_n^k, Y_n, X^1, \ldots, X^e]$, the causal graph $\mathcal{G}$, bound on the number of latent variables $l$.

1: $\Sigma_n \leftarrow \mathbf{c}^{(2)}(\mathbf{V}_n)$ {Sample covariance matrix}
2: **for all** $j \in [s]$ **do**
3:      $\mathrm{ad}(I^j, Y) \leftarrow \mathrm{an}(I^j) \cap \mathrm{an}(Y) \cap \mathcal{O}$ {Valid adjustment set}
4:      $\mathbf{b}_{Y,I^j,n} \leftarrow (\Sigma_n)_{Y,I^j|\mathrm{ad}(I^j,Y)}/(\Sigma_n)_{I,I^j|\mathrm{ad}(I^j,Y)}$ {Regression adjustment (Henckel et al., 2022, Prop. 1)}
5: **end for**
6: $Y_n^I \leftarrow Y_n - \sum_{j\in[s]} \mathbf{b}_{Y,I^j,n} I_n^j$ {(20)}
7: **for all** $i \in [k]$ **do**
8:      $T_n^{I,i} \leftarrow T_n^i$
9:      **for all** $j \in [s]$ **do**
10:          **if** $I_j$ is a valid instrument for $T_k$ in $\mathcal{G}$ **then**
11:              $\mathrm{ad}(I^j, T^i) \leftarrow \mathrm{an}(I^j) \cap \mathrm{an}(T^i) \cap \mathcal{O}$
12:              $\mathbf{b}_{T^i,I^j,n} \leftarrow (\Sigma_n)_{T^i,I^j|\mathrm{ad}(I^j,T^i)}/(\Sigma_n)_{I,I^j|\mathrm{ad}(I^j,T^i)}$
13:              $T_n^{I,i} \leftarrow T_n^i - \mathbf{b}_{T^i,I^j,n} I_n^j$ {(20)}
14:          **end if**
15:      **end for**
16:      $\mathbf{b}_n^i \leftarrow$ roots of $p_{[T_n^{I,i}, Y_n^I], l}(\mathbf{b}) = 0$ {(18)}
17: **end for**
18: $d_{\min} \leftarrow \infty$
19: **for all** $\mathbf{b}_n \in \mathbf{b}_n^1 \times \cdots \times \mathbf{b}_n^k$ **do**
20:      $d_b \leftarrow \min_{\mathbf{b}\in\mathcal{V}_I} ||\mathbf{b} - \mathbf{b}_n||_2^2$ {(19)}
21:      **if** $d_b < d_{\min}$ **then**
22:          $d_{\min} \leftarrow d_b$
23:          $\mathbf{b}_{Y,T^1,\ldots,T^k}^n \leftarrow \arg\min_{\mathbf{b}\in\mathcal{V}_I} ||\mathbf{b} - \mathbf{b}_n||_2^2$
24:      **end if**
25: **end for**
26: **RETURN:** $\mathbf{b}_{Y,T^1,\ldots,T^k}^n$

*Table 1.* Summary of the experimental setups.

| Figure | Causal Graph | Distribution | | | Parameters of Interest |
|---|---|---|---|---|---|
| | | **Family** | shape | scale | |
| 6 (left) | $\mathcal{G}_1$ in Fig. 5 | Gamma | U(0.1, 1) | U(0.1, 0.5) | $T \to Y$ |
| 6 (middle) | $\mathcal{G}_2$ in Fig. 5 | Gamma | U(0.1, 1) | U(0.1, 0.5) | $T \to Y$ |
| 6 (right) | $\mathcal{G}_3$ in Fig. 5 | Gamma | U(0.1, 1) | U(0.1, 0.5) | $T \to Y$ |
| 10 (left) | Fig. 9 | Gamma | U(0.1, 1) | U(0.1, 0.5) | $T \to Y$ |
| 7 | Fig. 4 | Gamma | U(0.1, 1) | U(0.1, 0.5) | $T_1 \to Y, T_2 \to Y$ |
| | | **Family** | alpha | beta | |
| 11 (left) | $\mathcal{G}_1$ in Fig. 5 | Beta | U(1.5, 2) | U(2, 10) | $T \to Y$ |
| 11 (middle) | $\mathcal{G}_2$ in Fig. 5 | Beta | U(1.5, 2) | U(2, 10) | $T \to Y$ |
| 11 (right) | $\mathcal{G}_3$ in Fig. 5 | Beta | U(1.5, 2) | U(2, 10) | $T \to Y$ |
| 10 (right) | Fig. 9 | Beta | U(1.5, 2) | U(2, 10) | $T \to Y$ |
| 12 | Fig. 4 | Beta | U(1.5, 2) | U(2, 10) | $T_1 \to Y, T_2 \to Y$ |

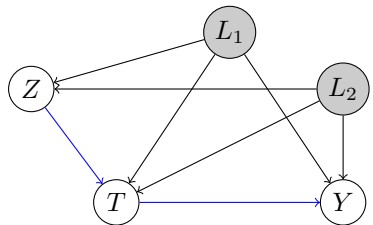

*Figure 9.* Proxy variable graph with an edge from proxy to treatment and two latent confounders.

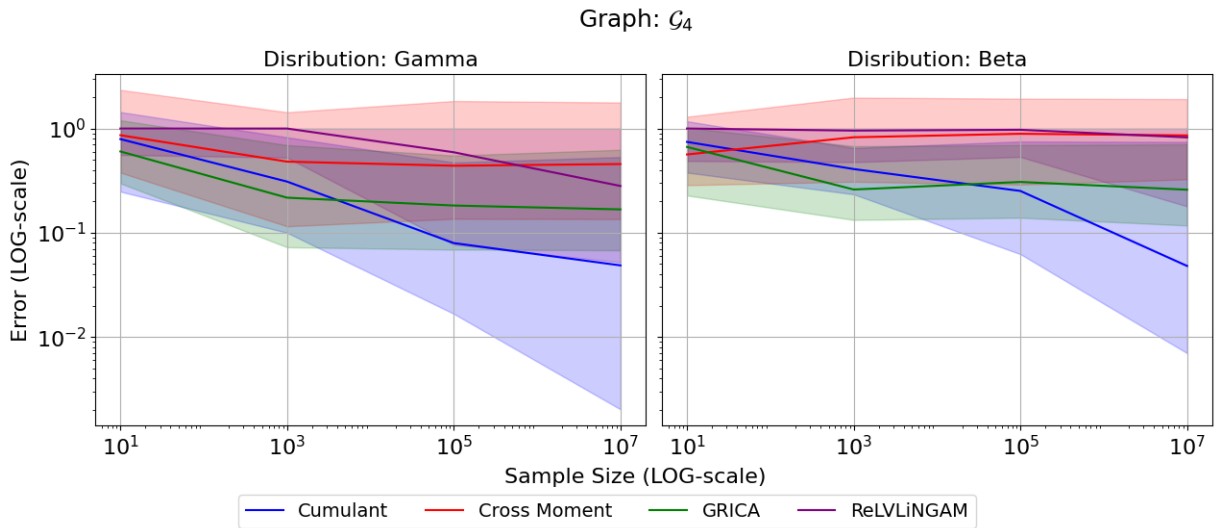

*Figure 10.* Relative error vs sample size for the graphs in Fig. 5.

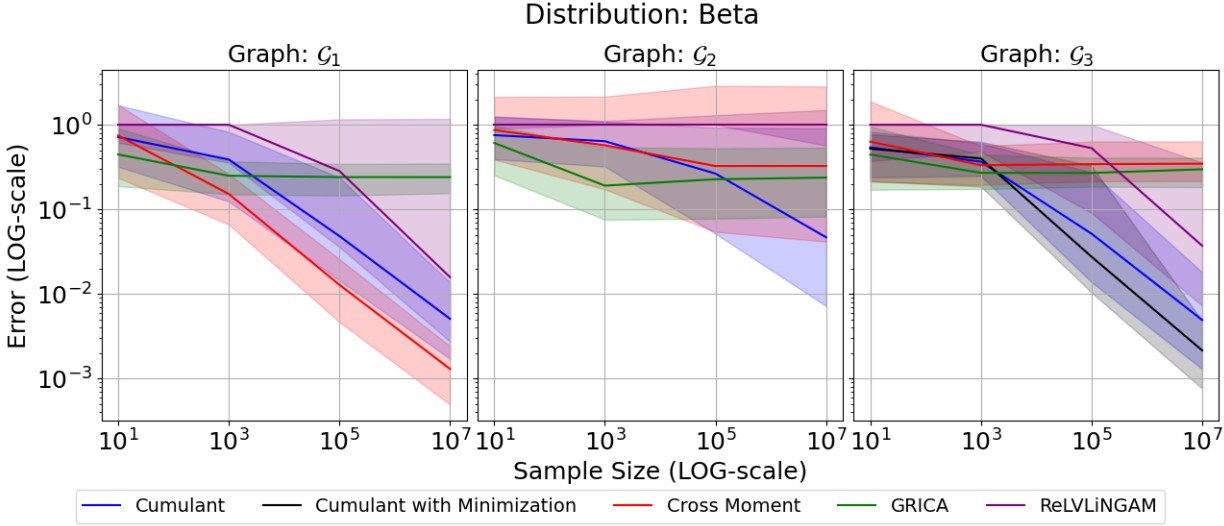

*Figure 11.* Relative error vs sample size for the graphs in Fig. 5.

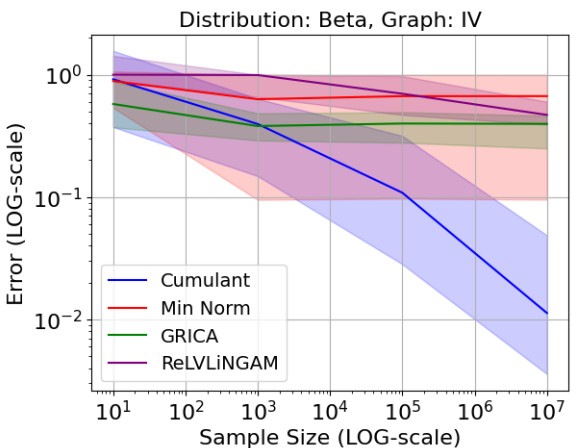

*Figure 12.* Relative error vs sample size for the graph in Fig. 4.

