# OpenReview forum: "Causal Effect Identification in lvLiNGAM from Higher-Order Cumulants"
_ICML.cc/2025/Conference — ICML 2025 poster_

### Official Review · Reviewer_nMo5 · 2025-03-09

**Overall Recommendation:** 4

**Summary:**

This paper propose causal effect identification method for proxy variable setup and underspecified instrumental variable setup based on high-order cumulants. In the proxy variable setup, both multiple latent confounders and a causal edge from the proxy variable to the treatment are allowed while only one proxy variable is required. In the underspecified instrumental variable setup, multiple treatments are allowed while only one instrumental variable is required.

**Claims And Evidence:**

The claims made in the submission are supported by evidence.

**Essential References Not Discussed:**

There is no related work that is essential to understanding the (context for) key contributions of the paper, but are not currently cited/discussed in the paper. But there are some recent works [1,2] that also use high-order cumulants for identification of LiNGAM with latent variables. I recommend the authors to discuss them.

[1] Identification of causal structure with latent variables based on higher order cumulants. AAAI 2024.

[2] Recovery of Causal Graph Involving Latent Variables via Homologous Surrogates. ICLR 2025.

**Experimental Designs Or Analyses:**

I have no concern about the soundness/validity of the experimental designs or analyses.

**Methods And Evaluation Criteria:**

The proposed methods and evaluation criteria make sense for the problem.

**Other Comments Or Suggestions:**

Personally, I think this paper is hard to follow because the theoretical tools used in this paper are not common in the community of causal inference. I fully acknowledge that this is not a deficiency per se, but this may limit the impact of this paper.

**Other Strengths And Weaknesses:**

# Strengths

1. This paper is well-motivated, Tramontano et al., (2024b) provides profound identifiability results, but their identification method is based on OICA, which has not yet been equipped with consistency guarantees. To overcome this limitation, this paper provides an identification method based on high-order cumulants.

2. This paper is solid, it provides both theoretical results (although the proofs of which are difficult to check for me) and experimental results.

# Weakness

1. Tramontano et al., (2024b) have already provided necessary and sufficient graphical conditions for  generic identifiability, so this paper is not novel in terms of identifiability results, its novelty lies in only the identification method.

2. (Minor) According to the experimental results shown as Figure 6, the proposed method is not superior to that proposed by Tramontano et al., (2024b) when the sample size is not very large.

**Questions For Authors:**

It seems that the proposed method assumes that the underlying causal graph is known. Please correct me if I'm wrong.

**Relation To Broader Scientific Literature:**

The authors have discussed this in Impact Statement.

**Theoretical Claims:**

I have tried my best to read the theoretical proofs. However, I'm not familiar with the concepts defined in Definition A.1~A.3. In fact, I have never encountered those concepts except for a recent work (Tramontano et al., 2024b). Besides, the proofs in this paper use theorems in (Schkoda et al., 2024), which are quite complicated. Therefore, I cannot provide a reliable assessment for the correctness of theoretical proofs in this paper.

---

> ### Author Rebuttal · Authors · 2025-03-31
>
> We thank the reviewer for their detailed comments and helpful suggestions.
>
> - *There is no related work that is essential to understanding the (context for) key contributions of the paper but are not currently cited/discussed in the paper. However, there are some recent works [1,2] that also use high-order cumulants for identification of LiNGAM with latent variables. I recommend the authors discuss them.*
>
> We will include a discussion of these works in the related work section. Both of these papers focus on the problem of *causal discovery*—that is, learning the causal structure from observational data—rather than on the problem of *causal effect identification*.  In the latter, the causal graph is assumed to be known, and the goal is to identify the causal effect of a treatment on an outcome (see, for example, the formal definition of this problem in the seminal work of Shpitser \& Pearl,  (2006).
>
> In particular, [1] proposed a procedure for using high-order moments to test for the presence of an edge between two observed variables where there is exactly one latent confounder. It is noteworthy that Schkoda et al. (2024) extended this result to settings with an arbitrary number of latent variables.
>
> In [2], the authors considered specific causal graphs under the assumption that for each latent variable $L$ in the graph, there exists a ``homologous surrogate" (see the definition in [2]). They then derived a set of structural properties using equations involving cumulants to recover the underlying causal structure. It is important to note that all the graphs considered in our work do not satisfy the assumption that every latent variable has a homologous surrogate except the graph $\mathcal{G}_1$, with a single latent variable and no edge from the proxy variable to the treatment.
>
> **Shpitser \& Pearl, (2006)** - Shpitser \& Pearl,  Identification of Joint Interventional Distributions in Recursive Semi-Markovian Causal Models, AAAI, 2006.
>
> - *It seems that the proposed method assumes that the underlying causal graph is known. Please correct me if I’m wrong.*
>
> As we mentioned above, in the problem of causal effect identification (the focus of the current work), the causal graph is known.
>
> - *Tramontano et al. (2024b) have already provided necessary and sufficient graphical conditions for generic identifiability, so this paper is not novel in terms of identifiability results; its novelty lies only in the identification method.*
>
> The identifiability results in Tramontano et al. (2024b) rely on the assumption that the full observed distribution is known. In contrast, our identifiability results require only knowledge of finitely many moments of the distributions, which is a strictly weaker assumption.
>
> Moreover, the estimation method proposed by Tramontano et al. (2024b) is based on solving an overcomplete Independent Component Analysis (OICA) problem, which is inherently non-separable. This implies that the true mixing matrix cannot be recovered solely by optimizing for independence among the exogenous noise—precisely the approach taken by their algorithm. Consequently, their method fails to consistently estimate the correct solution. This limitation is evident in our experiments, where we observe that GRICA’s accuracy does not improve monotonically with sample size.
>
> - *(Minor) According to the experimental results shown in Figure 6, the proposed method is not superior to that proposed by Tramontano et al. (2024b) when the sample size is not very large.*
>
> One possible explanation is that cumulant-based methods process unbiased estimates of high-order cumulants (order 4 or higher), also known as k-statistics. While these estimators are unbiased, they tend to have high variance for small sample sizes.
>
> In contrast, GRICA solves an optimization problem involving the $\ell_1$-norm of the observed samples, which have lower sample variance. As a result, for small sample sizes, GRICA may yield a lower mean squared error due to reduced variance. However, since the solution obtained by GRICA is not asymptotically unbiased, it cannot provide an asymptotically consistent estimator—unlike our proposed method.
>
> We will add a remark on this point in the final version of the manuscript.

---

> > ### Comment · Reviewer_nMo5 · 2025-04-02
> >
> > I thank the authors for their rebuttal. I have also read the authors' rebuttal to other reviews. I still have some concerns as follows.
> >
> > 1. In the problem of causal effect identification, the causal graph is **not** always assumed to be known. For instance, Tramontano et al. (2024b) have proven identifiability of causal effects in the case where the causal graph is unknown and the case there the causal graph is known separately. Therefore, I don't think the key distinction between causal discovery and causal effect identification lies in whether the causal graph is known.
> >
> > 2. Although Schkoda et al. (2024) focus on the problem of causal discovery, they also estimate the causal effects. Specifically, line 6 and 7 of Algorithm 1 in Schkoda et al. (2024) estimate both causal effects and cumulants of exogenous noise. This is very similar to Algorithm 1 in this paper. According to my understanding, Algorithm 1 in Schkoda et al. (2024) can identify causal effects even when the causal graph is unknown while Algorithm 1 in this paper can only identify causal effects when the causal graph is known. Please correct me if I'm wrong.
> >
> > 3. In line 233~234, the authors claim that " the cumulants of different exogenous noises are generically distinct". I think the authors may implicitly assume that the distributions of exogenous noises are not symmetric. As we know, for any random variable with symmetric distribution and any odd number $k$, $k$-th cumulant of the random variable is 0.
> >
> > I'm happy to raise my score if the authors can address the above concerns.
> >
> >
> > ===========After further rebuttal============
> >
> >
> > Most of my concerns have been addressed. As I promise, I increase my score to 4.

---

> > > ### Author Response · Authors · 2025-04-03
> > >
> > > - *In the problem of causal effect identification, the causal graph is not always assumed to be known. For instance, Tramontano et al. (2024b) have proven identifiability of causal effects in the case where the causal graph is unknown and the case there the causal graph is known separately. Therefore, I don't think the key distinction between causal discovery and causal effect identification lies in whether the causal graph is known.*
> > >
> > > Yes, indeed, there has been recent work on the joint identification of the graph and the causal effect. While this is certainly an interesting line of research, there is a much wider and well-established literature that considers the two problems as separate problems. In this work, we follow this latter tradition and assume that the causal graph is known.  It is important to note that the a priori knowledge of the causal graph leads to simpler identification procedures and, consequently, more efficient estimators (that are also more easily applied to real data).
> > >
> > > To make our point in an oversimplified setting:  Suppose we know that $X$ causes $Y$ and that no latent variables are at play. Then the causal effect of $X$ on $Y$ may be estimated by standard least squares regression. This is in contrast to more involved procedures that first need to resolve the causal direction and rule out the presence of latent variables.
> > >
> > > - *Although Schkoda et al. (2024) focus on the problem of causal discovery, they also estimate the causal effects. Specifically, line 6 and 7 of Algorithm 1 in Schkoda et al. (2024) estimate both causal effects and cumulants of exogenous noise. This is very similar to Algorithm 1 in this paper. According to my understanding, Algorithm 1 in Schkoda et al. (2024) can identify causal effects even when the causal graph is unknown while Algorithm 1 in this paper can only identify causal effects when the causal graph is known. Please correct me if I'm wrong.*
> > >
> > > It is true that when the causal effect is identifiable without knowledge of the graph, i.e., when Theorem 3.3 in Tramontano et al. (2024b) applies, Algorithm 1 in Schkoda et al. (2024) also identifies the correct causal effect. However, there are two issues with this approach:
> > >
> > > 1. There is a loss of statistical efficiency in jointly estimating the graph and the effect, as is apparent in Fig. 6. (This was also our point above).
> > >
> > > 2. There are instances in which the causal effects of interest are not identifiable without knowledge of the graph, such as the underspecified instrumental variable graph we consider in Section 3.2. In this case, an approach that does not explicitly impose causal assumptions on the graph would fail to identify the correct causal effect. This can be verified both theoretically (using Theorem 3.3 in Tramontano et al. (2024b)) and empirically through our experiments in Section 6.2.
> > >
> > > - *In line 233~234, the authors claim that "the cumulants of different exogenous noises are generically distinct". ... As we know, for any random variable with symmetric distribution and any odd number-th cumulant of the random variable is 0.*
> > >
> > > It is important to emphasize that the identifiability results in our work are all *generic*. Simply put, our statements hold for randomly sampled cumulant tensors (see Sec. 2.3 for a formal definition). This framework is arguably the most natural and compelling for studying causal effect identification in linear structural equation models, and it is indeed the approach used in all the papers mentioned above. In contrast, *global* identifiability—i.e., identifiability that holds for all cumulant tensors—has already been fully characterized (see Drton et al. (2011)) and is too restrictive to accommodate many relevant scenarios, such as Instrumental Variable regression. For a detailed discussion on the distinction between global and generic identifiability in linear structural equation models, we refer to Part III of Drton (2018).
> > >
> > > Regarding the reviewer's statement, it is indeed true that if the exogenous noise distributions are symmetric, their $(2k-1)$-th cumulants vanish, making them indistinguishable based on odd-order cumulants. However, this issue is already accounted for in our notion of genericity. Specifically, for any $k$, the set of cumulant tensors corresponding to symmetric distributions forms a measure-zero subset of $\mathcal{M}^{(\leq k)}(\mathcal{G})$. In other words, cumulant tensors associated with symmetric distributions are not generic.
> > >
> > > Practically, even when restricting to symmetric distributions, our approach remains valid by considering only even-order cumulants. This comes at the cost of using higher-degree cumulants than $k(l)$, but the core identifiability argument remains intact.
> > >
> > > **Drton et al. (2011)** - M. Drton, R. Foygel, and S. Sullivant, Global identifiability of linear structural equation models, 2011.
> > >
> > > **Drton, (2018)** - M. Drton, Algebraic problems in structural equation modeling, 2018.

---

### Official Review · Reviewer_oHRC · 2025-03-12

**Overall Recommendation:** 3

**Summary:**

The paper studies the problem of estimating causal effects in lvLiNGAM via higher-order cumulants. Specifically, the authors consider two setups where a single proxy variable exists and a instrumental variable (IV) exists with multiple treatments. In both settings, the authors provide the effect identification results and corresponding estimation methods. The conducted experiments verify the effectiveness of proposed methods.

**Claims And Evidence:**

The claims are generally well-supported by theoretical analysis.

**Essential References Not Discussed:**

NAN

**Experimental Designs Or Analyses:**

The experimental design and analyses are generally sound.

**Methods And Evaluation Criteria:**

The proposed method is reasonable and well-motivated.

**Other Comments Or Suggestions:**

Typo:

line 131: $\mathcal{M}(G))$ should be $\mathcal{M}(G)$

line 153: contain should be contains

line 665: fist should be first

**Other Strengths And Weaknesses:**

strengths

1. The paper is well-written.

2. This paper extends the original results of [1] that can only identify the effects up to some equivalence values. With the help of additional variables, proxy or IV, this paper shows the effect is identifiable.



Weaknesses or Questions

1. Lack of intuitive example or explanations for the main theorems. It could be better to remove the proofs in the main text into appendix and add more discussion.

2. What is the key role of the addition variables (proxy and IV) compared with the results in [1]? How does it benefit the identification?

3. Why does the method of 'Cumulant with Minimization' only occur in the experiments regarding the graph $\mathcal{G}_3$.

Reference:

[1] Schkoda, D., Robeva, E., and Drton, M. Causal discovery of linear non-gaussian causal models with unobserved confounding.

**Questions For Authors:**

See Weakness.

**Relation To Broader Scientific Literature:**

NAN

**Theoretical Claims:**

The theoretical claims are clearly presented, and the proofs are well-structured.

---

> ### Author Rebuttal · Authors · 2025-03-31
>
> We thank the reviewer for their detailed comments and helpful suggestions.
>
> - *Lack of intuitive example or explanations for the main theorems. It could be better to
> move the proofs in the main text into the appendix and add more discussion.*
>
> We will use the additional page in the final version of the paper to provide further explanations and enhance the clarity of our theorems.
>
> In short, all our theorems are based on the following intuition. Theorem 3.1 establishes that there exists a polynomial of degree $l+1$ (where $ l $ is the number of latent variables in Fig. 1) whose coefficients can be computed using only the cumulants up to order $k(l)$. The roots of this polynomial correspond exactly to the causal effects of the variables $L_i$ for $1\leq i \leq l $ and $ V_1 $ on $ V_2 $. Solving this polynomial system reduces the space of possible solutions for the causal effect from $ V_1 $ to $ V_2 $ to a finite set of size $ l+1$.
>
> To further refine this solution and identify the correct causal effect uniquely, we need to derive additional equations that can only be satisfied by the true causal effect. This requires analyzing the nonlinear relationships among the cumulants of the observed distribution. Theorems 3.4--3.7 detail how this approach applies to the respective graphs.
>
> - *What is the key role of the additional variables (proxy and IV) compared with the results in [1]? How does it benefit the identification?*
>
> The problem addressed in [1] pertains to *causal discovery*—learning the causal structure from observational data—which is distinct from the problem of causal effect identification. Specifically, [1] seeks to recover all causal graphs consistent with the observed data, but multiple such graphs may exist, leading to different possible causal effects. In contrast, causal effect identification assumes that the causal structure is known and focuses on determining the effect of a treatment on an outcome. This distinction is fundamental in the literature (see, for example, the seminal work by Shpitser \& Pearl, (2006). Following this framework, we assume that the causal graph is given and aim to uniquely identify the target causal effect.
>
> **Shpitser \& Pearl, (2006)** - Shpitser \& Pearl,  Identification of Joint Interventional Distributions in Recursive Semi-Markovian Causal Models, AAAI, 2006.
>
> - *Why does the method of 'Cumulant with Minimization' only occur in the experiments regarding the graph?*
>
> As mentioned in the final paragraph of Page 6, for the proxy setup shown in Figure 3 with a single latent variable (i.e., graph $\mathcal{G}_3$), we proposed an optimization-based technique that relies on computing lower-order cumulants compared to those used in the standard "Cumulant" method. We refer to this approach as ``Cumulant with Minimization", and it is tailored specifically to the structure of $\mathcal{G}_3$. As such, its results are presented only for graph $\mathcal{G}_3$ in Figure 6.

---

> > ### Comment · Reviewer_oHRC · 2025-04-04
> >
> > Thank you for the author's response. Most of my questions are addressed.
> >
> >
> > I will keep my score leaning towards acceptance.

---

### Official Review · Reviewer_6fLa · 2025-03-14

**Overall Recommendation:** 3

**Summary:**

This paper explores causal effect identification in latent variable Linear Non-Gaussian Acyclic Models (lvLiNGAM) using higher-order cumulants, addressing two challenging scenarios involving latent confounding: (1) a single proxy variable that may influence the treatment and (2) underspecified instrumental variable (IV) cases with fewer instruments than treatments. The authors theoretically prove that causal effects are identifiable under these conditions and propose corresponding closed-form estimation methods. Experimental results demonstrate the accuracy and robustness of the proposed approaches.

## update after rebuttal

I thank the authors for their rebuttal. My score remains unchanged.

**Claims And Evidence:**

The claims in this paper are clear and convincingly supported.

**Essential References Not Discussed:**

I didn't find other essential references that are not discussed.

**Experimental Designs Or Analyses:**

This paper evaluates its proposed methods through experiments on synthetic data across several representative causal graphs. For each of the two settings, the authors compare their approach against baseline methods, with results demonstrating its advantages.

While the findings support the effectiveness of the proposed methods, applying them to a real-world dataset may further strengthen the validation and highlight their practical applicability.

**Methods And Evaluation Criteria:**

This paper builds on the existing literature on LiNGAM-based structure discovery, shifting focus toward causal effect identification, which has received comparatively less attention. Following the lvLiNGAM framework, the study aims to identify specific entries of the mixing matrix using finitely many cumulants of the observational distribution under two challenging settings.

By leveraging higher-order cumulants, the proposed method introduces the following advancements:
a). Single Proxy Variable Setting: It allows a causal edge from the proxy to the treatment, thereby relaxing the previous assumption that each latent confounder must have exactly one proxy variable.
b). Underspecified Instrumental Variable Setting: It relaxes the assumption that the number of instrumental variables must be at least equal to the number of treatments.

The proposed approach appears methodologically sound, with a well-justified use of cumulants for causal effect identification.

**Other Comments Or Suggestions:**

All are listed above.

**Other Strengths And Weaknesses:**

All are listed above.

**Questions For Authors:**

All are listed above.

**Relation To Broader Scientific Literature:**

This paper focuses on relaxing assumptions and enhancing identifiability in causal effect estimation, which contributes to the broader literature on causal inference. By improving identifiability, the proposed approach may increase the applicability of causal effect estimation methods to real-world problems, making them more practical in settings where strict assumptions are difficult to satisfy.

**Theoretical Claims:**

I have looked through the theorems and proofs and did not find evident issues. However, as I am not an expert in the causal effect identification domain, I will defer to other reviewers for further validation during the rebuttal phase.

---

> ### Author Rebuttal · Authors · 2025-03-31
>
> We thank the reviewer for their detailed comments and helpful suggestions.
>
> - *While the findings support the effectiveness of the proposed methods, applying them to a real-world dataset may further strengthen the validation and highlight their practical applicability.*
>
>
> Following the input of the reviewer, we have evaluated our methods on the data from Card and Krueger (1993), similar to the experiments conducted by Kivva et al. (2023). We provide a brief overview of the results here, while a more complete discussion will be included in the revised version of the manuscript.
>
> The goal of the study is to estimate the effect of the minimum wage on the employment rate. In our experiments, we utilized the same preprocessing procedure in Kivva et al, (2023) and found that the results of our method, when assuming graphs $\mathcal{G}_1$ or $\mathcal{G}_2$, are consistent with findings in the existing literature. Specifically, the estimated causal effects under $\mathcal{G}_1$ and $\mathcal{G}_2$ are 2.68 and 2.71, respectively. Previous methods, such as the cross-moment approach (Kivva et al., 2023) and the Difference-in-Differences method, also provide an estimate of 2.68.
> In contrast, when assuming $\mathcal{G}_3$ as the true graph, we estimate a causal effect of 8.26. While this result still indicates a positive effect of the treatment on the outcome—consistent with prior work—the value of the estimated causal effect diverges from the ones reported in the literature. This suggests that, for this dataset, a causal graph excluding an edge from the proxy to the treatment (as in $\mathcal{G}_1$ or $\mathcal{G}_2$) may provide a better representation of causal relationships.
>
> The code to reproduce the results can be found at https://anonymous.4open.science/r/CEId-from-Moments-20AC/estimation/real_data.ipynb.
>
> **Card and Krueger (1993)** - D. Card and A. B. Krueger. Minimum wages and employment: A case study of the fast food industry
> in new jersey and pennsylvania, 1993.
>
> **Kivva et al., (2023)** - Y. Kivva, S. Salehkaleybar, N. Kiyavash, A Cross-Moment Approach for Causal Effect Estimation, NeuriPS 2023.

---

### Decision · Program_Chairs · 2025-05-01

**Decision:**

Accept (poster)

**Comment:**

This paper addresses the challenging problem of causal effect identification in latent variable linear non-Gaussian acyclic models (lvLiNGAM), focusing on two realistic and underexplored settings: identification using a single proxy variable, and underspecified instrumental variable scenarios. By leveraging higher-order cumulants, the authors establish identifiability results and propose corresponding estimation methods, supported by both theoretical analysis and empirical evaluation.

The reviewers agree that the paper is well-motivated, clearly written, and contributes meaningfully to the literature on causal inference with latent confounding. Reviewer 6fLa and oHRC appreciate the extension beyond existing LiNGAM frameworks and confirm the soundness of the theoretical and empirical results. Reviewer nMo5 notes that while some core identifiability conditions have been addressed in prior work (e.g., Tramontano et al., 2024b), the novelty of this paper lies in proposing a tractable, cumulant-based estimation method with clearer guarantees. Some concerns were raised about the complexity and accessibility of the theoretical tools, and the experiments are currently limited to synthetic data. A real-world application could further strengthen the practical relevance in future work. Overall, I recommend acceptance if there is room.